# VLM-RobustBench: A Comprehensive Benchmark for Robustness of Vision-Language Models

**Rohit Saxena** [1]  **Alessandro Suglia** [1]  **Pasquale Minervini** [1 2]

## Abstract

Vision-language models (VLMs) achieve strong performance on standard, high-quality datasets, but we still do not fully understand how they perform under real-world image distortions. We present **VLM-RobustBench**, a benchmark spanning 49 augmentation types across noise, blur, weather, digital, and geometric perturbations, evaluated under graded severities (low/mid/high) and binary transforms, yielding 133 corrupted settings. We evaluate 15 VLMs from five families, including proprietary GPT-5.4-mini, on two complementary benchmarks: MMBench (visually grounded) and MMMU-Pro (reasoning-oriented). Our results reveal that visual severity is a weak predictor of difficulty: low-severity spatial perturbations often degrade performance more than visually severe photometric corruptions. In particular, low-severity `glass_blur` reduces MM-Bench accuracy by about 8 pp on average across models, while the largest drops arise from resampling and geometric distortions (e.g., `upsample`, `elastic_transform`), reaching up to 34 pp. Overall, our findings suggest current VLMs are *semantically strong but spatially fragile*, motivating the definition of novel robustness evaluation protocols and training regimes that emphasize resampling and geometric invariances.

## 1. Introduction

The rapid evolution of Vision-Language Models (VLMs) has marked a transition from text-only specialists to multimodal generalist models that are capable of complex reasoning across a broad range of tasks. Recent VLMs (Bai et al., 2025; Clark et al., 2026; Wang et al., 2025a; Team et al., 2025) demonstrated remarkable proficiency on sev-

eral benchmarks, exhibiting capabilities ranging from zero-shot generalisation to fine-grained image-text understanding (Liu et al., 2024). These advancements catalysed the integration of VLMs into safety-critical pipelines, including autonomous driving perception stacks (Zhou et al., 2024), robotics (Intelligence et al., 2026), medical diagnostic support systems (Sellergren et al., 2025), and automated document processing workflows (Wang et al., 2025b).

However, strong performance on curated benchmarks does not guarantee reliability under the distribution shifts encountered in deployment (Yu et al., 2024). Real-world visual inputs are rarely pristine: low-light sensor noise, adverse weather (rain, fog, snow), compression artefacts, and motion or defocus blur are common (Hendrycks & Dietterich, 2019). In addition, viewpoint changes induce geometric variations, such as scaling, rotation, and perspective distortion, that may be absent or simplified in training data (Barbu et al., 2019b; Zhou et al., 2022). For safety-critical use, we therefore need robustness evaluations that stress these everyday corruptions rather than measuring accuracy on a fixed dataset that includes only a few visual perturbations.

While the computer vision community has established rigorous benchmarks for robustness to common corruptions—most notably ImageNet-C (Hendrycks & Dietterich, 2019), the robustness landscape for modern VLMs remains less systematically characterised, especially across tasks and realistic corruption families (Usama et al., 2025; Ye et al., 2024). A central challenge is understanding whether language-side reasoning can compensate when visual perception is degraded, or whether certain perturbations induce sharp perceptual bottlenecks that dominate end performance (Liu et al., 2025; Fan et al., 2025; Zhou et al., 2025).

Moreover, corruption benchmarks often implicitly assume *severity monotonicity*: as visual distortion increases, inputs should become increasingly harder (Hendrycks & Dietterich, 2019). It remains unclear whether this assumption holds for VLMs, where perception and language reasoning are tightly coupled through cross-modal representations (Zhou et al., 2025). This motivates a dedicated benchmark that probes the interplay between visual corruptions and multimodal reasoning across a broad spectrum of perturbation types and

---

[1]University of Edinburgh, UK [2]Miniml.AI. Correspondence to: Rohit Saxena <rohit.saxena@ed.ac.uk>.

*Proceedings of the 43rd International Conference on Machine Learning*, Seoul, South Korea. PMLR 306, 2026. Copyright 2026 by the author(s).

severity levels.

In this work, we present **VLM-RobustBench**[1], a large-scale analysis of VLM robustness under visual corruption. We systematically evaluate 15 models spanning five families (Qwen3-VL, InternVL3.5, Molmo2, Gemma 3, and proprietary GPT-5.4-mini) with scales from 4B to 235B parameters, across 133 distinct augmentation configurations (42 corruptions at three severities plus 7 binary transforms) on two diverse datasets: MMBench (Liu et al., 2024) (more visually grounded) and MMMU-Pro (Yue et al., 2025) (more reasoning-oriented). Our results challenge prevailing assumptions and reveal that current VLMs are *semantically strong but spatially fragile*. We highlight three key contributions:

1. **The Spatial Fragility Finding:** VLMs are disproportionately sensitive to spatial and resampling artefacts. A resampling operation (`upsample`) or geometric distortion causes catastrophic failure (up to 34pp drop), whereas severe photometric degradations (e.g., noise, compression) are often handled robustly.
2. **Severity Mismatch:** We observe a decoupling of severity level and model difficulty (Figure 1). On MMBench, low-severity perturbations degrade performance more than high-severity perturbations of other types, complicating safety assurance (e.g., `glass_blur` and `solarize` at low severity result in a 7.9pp and 5.7pp drop, respectively).
3. **Family-Specific Vulnerabilities:** Robustness is not a function of parameter count. Distinct model families exhibit unique vulnerability "fingerprints," suggesting that architectural choices and training regimes play a decisive role in determining failure modes.

## 2. Related Work

**Vision–language model evaluation benchmarks.** The rapid progress of vision–language models (VLMs) has driven a parallel effort on standardised evaluation. Widely used benchmarks measure complementary aspects of multimodal capability, including broad multi-skill perception and reasoning (Liu et al., 2024), discipline-spanning expert knowledge and reasoning (MMMU and its variant MMMU-Pro) (Yue et al., 2024; 2025), and structured decompositions of perception versus cognition (Fu et al., 2025). Beyond aggregate scores, recent work highlights that some VLMs can answer via language priors with limited visual grounding (Tong et al., 2024), motivating "vision-centric" evaluations such as NaturalBench (Li et al., 2024) that reduce "blind" shortcuts, and comparison-consistency benchmarks (Feizi et al., 2025). Our work builds on this eval-

[1]Code: https://github.com/saxenarohit/vlm_robustbench

uation ecosystem but focuses on reliability under visual degradations, using MMBench (more visually grounded) and MMMU-Pro (more reasoning-oriented) to ensure a robust assessment. Additionally, we use *visual gain* to quantify directly reliance on visual information versus language priors.

**Robustness to natural corruptions in vision.** Robustness to common, naturally occurring corruptions has a long history in computer vision, formalised by benchmarks such as ImageNet-C/ImageNet-P (Hendrycks & Dietterich, 2019), which apply parameterised families of noise, blur, weather, digital, and geometric perturbations with calibrated severities. Related benchmarks extend distribution shift beyond corruptions, including style/rendition shifts (ImageNet-R) (Hendrycks et al., 2021) and viewpoint/background changes (ObjectNet) (Barbu et al., 2019a). VLM-RobustBench follows the corruption-benchmark philosophy but adapts it to VLM settings, expanding the augmentation taxonomy and explicitly separating graded severities from binary transforms to reflect real deployment conditions (e.g., resampling artefacts, watermarks, borders).

**Natural-corruption robustness in VLMs and VQA.** Compared to vision-only models, the robustness of VLMs under visual corruption is less mature. Recent studies evaluate VLMs under subsets of ImageNet-C-like corruptions and introduce corrupted VQA settings, highlighting sensitivity patterns that differ across tasks (Usama et al., 2025). Our benchmark differs in (i) breadth of corruptions (including resampling and geometry-focused stressors and VLM-specific artefacts), (ii) explicit severity analysis (low/mid/high) alongside binary transforms, and (iii) evaluation on both visually grounded and reasoning-oriented multimodal benchmarks, enabling analysis of when models fall back to language priors versus visual grounding.

**Adversarial robustness of vision–language models.** A distinct line of work studies worst-case, adversarial perturbations for vision–language pretraining models and VLMs, including transferable and black-box attacks (Zhao et al., 2023; Qi et al., 2024; Shayegani et al., 2024), as well as defences that robustify the vision encoder or the multimodal alignment. Canonical demonstrations show that adversarial images can circumvent safeguards and induce incorrect or unsafe generations in multimodal models (Carlini et al., 2023). On the defence side, adversarially fine-tuning CLIP-style encoders (e.g., Robust CLIP) can improve robustness for downstream VLMs that rely on frozen vision backbones (Mao et al., 2023). While adversarial robustness is crucial for security, our focus is complementary: we target naturally occurring corruptions and operational artifacts that arise in the absence of an adaptive attacker, and we show that these "benign" perturbations can dominate tail risk.

| Original | Brightness (high) | Glass Blur (low) |
|---|---|---|

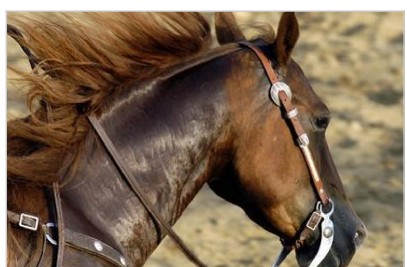 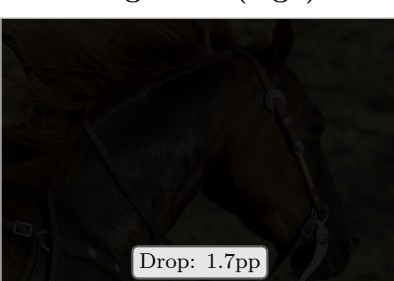 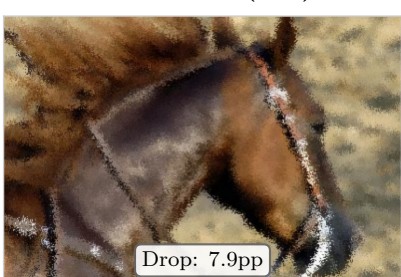

*Figure 1.* **The Severity Paradox.** On MMBench (mean over 12 open-weights models), high-severity brightness reduction (center) causes only a 1.7pp accuracy drop, while low-severity glass blur (right) causes a 7.9pp drop. Severity level does not always predict model difficulty.

**Spatial robustness and resolution effects.** Our main finding, high sensitivity to spatial/resampling corruptions, connects to robustness properties of patch-based encoders. Prior work analyses robustness of Vision Transformers under corruptions and distribution shift, motivating architectural and training choices for improved invariances (Bhojanapalli et al., 2021; Paul & Chen, 2022). In the VLM context, preprocessing choices (e.g., resizing strategy, tokenization granularity) can materially affect performance (McKinzie et al., 2024). VLM-RobustBench provides a systematic corruption suite to quantify these effects at scale and to guide robustness-oriented training curricula that emphasize geometric and resampling invariances.

## 3. Method

### 3.1. Problem Formulation

Let $\mathcal{M}$ denote a vision-language model that takes an image $I \in \mathcal{I}$ from the space of images $\mathcal{I} = \mathbb{R}^{H \times W \times 3}$ and a text query $Q$ as input, producing a textual response $\mathcal{M}(I, Q)$. We evaluate multiple-choice accuracy using an answer extractor $g(\cdot)$ that maps the response to a discrete option, yielding $\hat{y} = g(\mathcal{M}(I, Q))$.

We define a set of image augmentations $\mathcal{A} = \{A_1, \ldots, A_K\}$. Each severity-based augmentation $A_k$ is a stochastic transformation parameterized by severity $s \in \mathcal{S}$ that maps an image $I \in \mathcal{I}$ to one of its augmented versions $\hat{I} \in \mathcal{I}$, with $\hat{I} \sim A_k(I, s)$. For reproducibility, we fix per-sample random seeds, yielding deterministic outputs for each (image, severity) pair. Binary augmentations are not parameterised by $s$ and apply directly as $A_k(I)$.

Given a dataset $\mathcal{D} = \{(I_i, Q_i, y_i)\}_{i=1}^{N}$ with ground-truth answers $y_i$, we define clean accuracy as:

$$\text{Acc}_{\text{clean}} = \frac{1}{N} \sum_{i=1}^{N} \mathbf{1}[g(\mathcal{M}(I_i, Q_i)) = y_i],$$

and accuracy under augmentation $A_k$ at severity $s$ as:

$$\text{Acc}_{A_k,s} = \frac{1}{N} \sum_{i=1}^{N} \mathbf{1}[g(\mathcal{M}(A_k(I_i, s), Q_i)) = y_i]. \quad (1)$$

The robustness drop $\Delta_{A_k,s} = \text{Acc}_{\text{clean}} - \text{Acc}_{A_k,s}$ quantifies the performance degradation in percentage points. We additionally evaluate a no-image baseline $\text{Acc}_{\varnothing}$ where images are removed.

**Visual Gain.** To quantify reliance on visual information versus language priors, we define *Visual Gain* (VG) as

$$\text{VG} = \text{Acc}_{\text{clean}} - \text{Acc}_{\varnothing}. \quad (2)$$

A larger VG indicates stronger dependence on visual input, whereas a low VG suggests greater solvability from language priors alone.

**Relative Corruption Error.** To normalize corruption impact by a model's visual reliance, we define

$$\text{RCE}_{A_k,s} = \frac{\Delta_{A_k,s}}{\text{VG}} \times 100\%. \quad (3)$$

All model–dataset pairs in our experiments have VG > 7, so division is well-defined. RCE=100% means the corruption removes all visual benefit; RCE>100% means performance becomes worse than the no-image baseline.

### 3.2. Augmentation Taxonomy

We construct a suite of 49 image augmentations motivated by real-world degradations. The suite comprises 42 severity-based corruptions grouped into nine categories and 7 binary transforms (Table 1). Each severity-based corruption is evaluated at three levels $s \in \{\text{low}, \text{mid}, \text{high}\}$, whereas binary transforms are applied once (no severity parameter).

For corruptions overlapping ImageNet-C (Hendrycks & Dietterich, 2019), we reuse the same corruption types but calibrate severity levels independently for VLM evaluation. For

| Category | N | Augmentations |
|---|---|---|
| Blur | 5 | Gaussian, motion, defocus, glass, zoom |
| Noise | 4 | Gaussian, shot, speckle, salt-pepper |
| Weather | 5 | Fog, frost, snow, rain, spatter |
| Digital | 2 | JPEG compression, pixelation |
| Geometric | 5 | Rotate, shear, affine, perspective_transform, elastic |
| Occlusion | 3 | Center, random, grid mask |
| Color/Tone | 10 | Brightness$^\pm$, contrast$^\pm$, saturation$^\pm$, gamma$^\pm$, hue shift, color jitter |
| Resolution | 5 | Downsample, upsample, sharpen, posterize, solarize |
| VLM-specific | 3 | Text overlay, watermark, border |
| Binary | 7 | Grayscale, invert, equalize, autocontrast, channel swap, flip (h/v) |

*Table 1.* Augmentation taxonomy. We group 42 severity-based corruptions into nine categories and evaluate each at low/mid/high; 7 binary transforms are applied once. Superscript $\pm$ denotes separate increase/decrease variants.

| Family | Model | Type |
|---|---|---|
| Qwen3-VL | Qwen3-VL-4B | Instruct |
| | Qwen3-VL-8B | Instruct |
| | Qwen3-VL-30B-A3B | MoE, 3B active |
| | Qwen3-VL-32B | Instruct |
| | Qwen3-VL-235B | MoE, 22B active |
| | Qwen3-VL-4B-Think | Thinking |
| | Qwen3-VL-8B-Think | Thinking |
| InternVL3.5 | InternVL3.5-4B | Instruct |
| | InternVL3.5-8B | Instruct |
| | InternVL3.5-14B | Instruct |
| | InternVL3.5-38B | Instruct |
| Molmo2 | Molmo2-4B | Instruct |
| | Molmo2-8B | Instruct |
| Gemma 3 | Gemma-3-12B-it | Instruct |
| GPT-5.4 | GPT-5.4-mini | Closed-Source |

*Table 2.* Family of VLMs evaluated. Aggregate analyses use the 12 open-weights instruct models; GPT-5.4-mini is reported separately (Section A.14). Qwen `Think` models are reported as a test-time compute ablation.

VLM-specific corruptions, we define monotonic, visually ordered severity schedules (full parameter schedules in Section C.5). Low severity corresponds to mild perturbations, while high severity corresponds to strongly degraded inputs. This yields 126 severity-based configurations ($42 \times 3$ levels) plus 7 binary transforms, resulting in *133 augmentation configurations* per model–dataset pair.

## 4. Experimental Setup

### 4.1. Models

We evaluate **15 VLMs** spanning five model families: four open-weights families, Qwen3-VL (Bai et al., 2025), InternVL3.5 (Wang et al., 2025a), Molmo2 (Clark et al., 2026), and Gemma3 (Team et al., 2025), and one proprietary model, GPT-5.4-mini (Table 2). These models use three vision encoder families (SigLIP2, InternViT, SigLIP) with four distinct preprocessing pipelines: native dynamic resolution (Qwen3-VL), tile-based 448×448 (InternVL3.5), fixed 896×896 with Pan & Scan (Gemma 3), and multi-crop 378×378 with overlapping tiles (Molmo2).

Our primary robustness comparisons focus on *12 open-weights instruction-following VLMs* plus one proprietary model (GPT-5.4-mini), all evaluated under a consistent direct-answer prompting protocol. We additionally include **2 Qwen3-VL** `Think` models (4B and 8B) as a *test-time compute ablation*. To isolate the role of reasoning at inference time, we compare chain-of-thought prompting against the `Think` variants [2].

---

[2] These results are reported in Section A.4 and are not included in the main aggregate robustness comparisons.

### 4.2. Datasets

We evaluate on two challenging multimodal benchmarks (with seed 42 for reproducibility):

**MMMU-Pro** A professional-level multimodal understanding benchmark covering subjects from STEM to humanities. We use the standard 10-option multiple choice variant, evaluating on a stratified 20% sample (324 samples) across 30 subject categories.

**MMBench** A comprehensive benchmark for multimodal perception and reasoning. We evaluate on the English development split using stratified 20% sampling (853 samples) to ensure category balance across all question types.

### 4.3. Evaluation Protocol

For each model-dataset pair, we evaluate on clean images, a no-image baseline (image removed), and all corrupted settings: 126 severity-based configurations (42 corruptions × 3 severity levels) plus 7 binary transforms, totaling *133 + 2 (baseline) evaluations* per pair. Corruptions are applied to images only; text prompts and answer formats are held fixed across conditions. We use stratified 20% subsampling above to keep the full corruption sweep tractable (135 settings per model-dataset pair) while preserving category balance.

Unless stated otherwise, we report results in direct mode (standard prompting). Our 15 models comprise 12 open-weights instruct checkpoints, one proprietary model (GPT-5.4-mini), and 2 Qwen `Think` variants. All aggregated analyses are computed over the 12 open-weights models. GPT-5.4-mini results are reported separately (Section A.14). Chain-of-thought (CoT) and thinking results are reported

**MMBench (Direct)**

| Model | Baseline↑ | Worst-Case↓ | Severe-Fail↓ | Worst@Low↓ | Benign@Low↑ | VG↑ | mRCE↓ |
|---|---|---|---|---|---|---|---|
| Qwen3-VL-4B | 88.4 | 26.3 | 3.8 | 7.0 | 88.1 | 48.2 | 3.9 |
| Qwen3-VL-8B | 90.2 | 30.2 | 5.3 | 8.3 | 73.8 | 48.2 | 5.2 |
| Qwen3-VL-30B | 90.7 | 29.4 | 3.8 | 6.1 | 88.1 | 47.7 | **3.5** |
| Qwen3-VL-32B | 92.4 | 32.4 | 6.0 | 8.1 | 81.0 | 49.2 | 5.4 |
| Qwen3-VL-235B | 93.4 | 30.7 | 4.5 | 6.3 | 81.0 | 50.2 | 4.5 |
| InternVL3.5-4B | 86.3 | 30.5 | 9.8 | 10.8 | 64.3 | 44.9 | 7.7 |
| InternVL3.5-8B | 89.1 | 31.7 | 8.3 | 9.6 | 71.4 | **50.8** | 6.5 |
| InternVL3.5-14B | 86.6 | 29.4 | 9.0 | 9.5 | 88.1 | 44.5 | 6.0 |
| InternVL3.5-38B | 90.5 | 27.2 | 3.8 | 6.1 | 85.7 | 46.8 | 4.5 |
| Molmo2-4B | 88.5 | 33.1 | 4.5 | 7.4 | 78.6 | 43.6 | 5.5 |
| Molmo2-8B | 88.4 | 33.9 | 4.5 | 6.3 | **90.5** | 48.4 | 4.4 |
| Gemma-3-12B | 85.3 | 32.1 | 8.3 | 10.7 | 69.0 | 44.1 | 6.4 |
| GPT-5.4-mini | 87.7 | 31.8 | 8.3 | 11.7 | 85.7 | 46.3 | 5.9 |

**MMMU-Pro (Direct)**

| Model | Baseline↑ | Worst-Case↓ | Severe-Fail↓ | Worst@Low↓ | Benign@Low↑ | VG↑ | mRCE↓ |
|---|---|---|---|---|---|---|---|
| Qwen3-VL-4B | 31.5 | 7.4 | 3.8 | 2.8 | **95.2** | 7.1 | −6.8 |
| Qwen3-VL-8B | 35.2 | 9.0 | 7.5 | 5.2 | 85.7 | 11.4 | 4.7 |
| Qwen3-VL-30B | 40.7 | 14.5 | 12.8 | 7.4 | 59.5 | **17.0** | 12.1 |
| Qwen3-VL-32B | 43.2 | 13.0 | 6.8 | 4.9 | 69.0 | 12.7 | 11.6 |
| Qwen3-VL-235B | 46.0 | 20.7 | 12.0 | 13.6 | 85.7 | 13.6 | 9.4 |
| InternVL3.5-4B | 37.3 | 11.1 | 14.3 | 5.6 | 76.2 | 12.3 | 11.6 |
| InternVL3.5-8B | 41.0 | 12.3 | 16.5 | 5.6 | 45.2 | 14.2 | 16.5 |
| InternVL3.5-14B | 42.0 | 9.6 | 9.0 | 5.9 | 85.7 | 15.1 | 5.2 |
| InternVL3.5-38B | 41.4 | 9.6 | 6.0 | 4.0 | 83.3 | 13.0 | 6.7 |
| Molmo2-4B | 31.8 | 5.6 | 3.8 | 4.3 | 92.9 | 12.0 | 4.9 |
| Molmo2-8B | 31.2 | 5.6 | 5.3 | 4.0 | 90.5 | 7.4 | 1.0 |
| Gemma-3-12B | 33.0 | 9.9 | 27.1 | 9.0 | 28.6 | 10.8 | 24.2 |
| GPT-5.4-mini | 39.2 | 13.0 | 12.8 | 9.6 | 73.8 | 12.7 | 12.6 |

*Table 3.* Main robustness summary (12 open-weights models + GPT-5.4-mini). Worst-Case and Severe-Fail are computed over all 133 configurations (126 severity-based + 7 binary). Worst@Low and Benign@Low are computed over the 42 low-severity configurations only (binary excluded). mRCE is the mean Relative Corruption Error over all $|\mathcal{C}| = 133$ configurations; negative values reflect low visual reliance rather than robustness (see Section 5.6). VG and RCE are defined in Section 3.1.

separately in Section A.4. **Metrics.** We report clean accuracy $\text{Acc}_{\text{clean}}$ as defined in Section 3.1. Since mean accuracy aggregates across augmentation types and severities, it can mask severity-specific and tail failures; we therefore focus on drop-based metrics that highlight failure modes. For each configuration $(A_k, s)$, we define the accuracy drop (in percentage points) as $\Delta_{A_k,s} = \text{Acc}_{\text{clean}} - \text{Acc}_{A_k,s}$ (binary transforms omit $s$).

We additionally report: (i) *Worst-Case Drop* $\max_{k,s} \Delta_{A_k,s}$, the maximum drop over all 133 configurations (126 severity-based + 7 binary); (ii) *Severe-Failure Rate*, the fraction of the same 133 configurations for which performance drops by more than a relative threshold, $\Delta_{A_k,s} > 0.1\,\text{Acc}_{\text{clean}}$. (iii) *Worst@Low* $\max_k \Delta_{A_k,\text{low}}$, the maximum drop over 42 severity-based corruptions at low severity (binary excluded); and (iv) *Benign@Low*, the fraction of these 42 low-severity corruptions with $\Delta \leq 1$. Additional implementation details are in Section C.

| Augmentation | MMBench | | MMMU-Pro | |
|---|---|---|---|---|
| | Drop | Tier | Drop | Tier |
| Autocontrast | 0.0 | Benign | −0.2 | Positive |
| Grayscale | 3.2 | Moderate | 0.1 | Benign |
| Channel Swap | 3.0 | Moderate | −0.1 | Positive |
| Equalize | 3.1 | Moderate | 0.4 | Benign |
| Horizontal Flip | 7.0 | Moderate | 3.6 | Moderate |
| **Vertical Flip** | **10.2** | **Catastrophic** | 4.8 | Moderate |
| **Invert** | **10.4** | **Catastrophic** | 1.3 | Mild |

*Table 4.* Binary augmentation drops (pp) averaged over 12 open-weights models. Vertical flip and color inversion cause catastrophic failures on MMBench ($\Delta > 10\,\text{pp}$); vertical flip exceeds the mean drop of 39 out of 42 high-severity corruptions.

## 5. Results

### 5.1. Tiered Robustness Overview

Table 3 provides a tiered snapshot of robustness in direct mode. We report $\text{Acc}_{\text{clean}}$ (*Baseline*) alongside drop-based

tail-risk summaries that capture how models fail across the full corruptions: *Worst-Case Drop*, *Severe-Failure Rate*, *Worst@Low*, and *Benign@Low*. These metrics are deployment-relevant because robustness risk is often dominated by a small number of failure-inducing transformations (e.g., resampling in a preprocessing pipeline), even when most corruptions are harmless. We additionally report *VG* (visual reliance) and *mRCE* (mean relative corruption error) to normalize corruption impact by how much each model benefits from vision (Section 3.1).

Two patterns stand out. First, *large failures are sparse but consequential*: on MMBench, many low-severity corruptions are benign (typically $\Delta \leq 1$ for about 65–93% of the 42 low-severity settings, depending on the model), yet a small subset produces sharp accuracy drops. This is reflected by the *Severe-Failure Rate*, which measures how often a model experiences a drop larger than $0.1\,\mathrm{Acc_{clean}}$ across the 133 configurations. For example, InternVL3.5-4B attains a severe-failure rate of 9.8% on MMBench, meaning 13 out of 133 corrupted settings exceed this threshold.

Second, *visually mild perturbations can still be high-risk*. *Worst@Low* shows that even at low severity, some corruptions cause substantial degradation (up to 10.8 pp on MMBench). In contrast, MMMU-Pro exhibits a wider spread in *Benign@Low* (roughly 30–95% of the 42 low-severity settings), consistent with varying degrees of visual reliance across models and tasks.

**Preempting the "destroyed image" intuition.** Although high-severity corruptions can visibly degrade inputs, our main takeaway is that perturbation type matters more than severity level: spatial perturbations (e.g., glass_blur and resampling artifacts) can be as harmful as, or more harmful than, strong photometric distortions (e.g., JPEG compression), even at lower severity settings.

**Scaling does not eliminate spatial fragility.** Qwen3-VL-235B achieves the best mCE (50.0), yet still drops 30.7 pp worst-case on MMBench. GPT-5.4-mini, despite being proprietary, exhibits the highest Worst@Low (11.7 pp) and highest Flip$^{+}$ rate (5.23%) among all models (Sections A.12 and A.14), demonstrating that neither scale nor proprietary training eliminates this vulnerability.

## 5.2. Binary Augmentations: Trivial Transforms, Large Failures

Beyond the 126 severity-based configurations, our 133-config benchmark includes 7 binary (on/off) augmentations. Table 4 reveals that two trivial transformations—vertical flip and color inversion—are *catastrophic* on MMBench despite requiring no learned parameters.

**Key insight.** Vertical flip is more harmful than 39 of 42 high-severity corruptions on MMBench, exceeded only by

upsample, elastic_transform, and zoom_blur— suggesting VLMs encode strong orientation priors. Color inversion causes catastrophic drops on MMBench (10.4pp) but only mild harm on MMMU-Pro (1.3pp), indicating perception tasks depend on absolute color relationships while reasoning does not.

**Perception vs. Reasoning via Visual Gain.** We analyze Visual Gain (VG = $\mathrm{Acc_{clean}} - \mathrm{Acc_{\varnothing}}$) as a proxy for reliance on visual input versus language priors. VG is computed per model and then averaged across the 12 open-weights models. MMBench has substantially larger VG (47.2 points) than MMMU-Pro (12.2 points), indicating that MMBench decisions depend more strongly on visual grounding, whereas MMMU-Pro permits greater fallback to language priors. This aligns with MMBench exhibiting larger worst-case drops and higher severe-failure rates (Table 3).

## 5.3. Which Corruptions Drive Risk?

Figure 2 shows the most harmful corruptions at each severity, averaged over 12 open-weights models. Per-model breakdowns are in Section A.8. Two consistent patterns emerge. First, *perturbation type dominates over severity level*: at *low* severity, glass_blur (7.9 drop on MMBench, 5.1 on MMMU-Pro) is among the top failures, exceeding most high-severity photometric corruptions. Second, *catastrophic risk concentrates in a small set of resampling and geometric corruptions*: upsample and elastic_transform dominate mid/high severities on MMBench, while zoom_blur becomes more prominent on MMMU-Pro.

**Glass Blur Anomaly.** Low-severity glass_blur outperforms many high-severity photometric corruptions (e.g., JPEG compression), providing a concrete example of the severity–difficulty mismatch. Interestingly, glass_blur exhibits non-monotonic behaviour, with low severity sometimes inducing larger drops than higher severities (Section A.9), further illustrating the decoupling of visual and model difficulty.

## 5.4. Quantifying Severity Mismatch

Severity mismatch can be quantified by checking whether performance degrades monotonically with increasing visual severity. For each model and severity-based corruption $A_k$, we compute: (i) A *monotonicity violation* indicator, set to 1 if $\Delta_{A_k,\mathrm{low}} > \Delta_{A_k,\mathrm{mid}}$ or $\Delta_{A_k,\mathrm{mid}} > \Delta_{A_k,\mathrm{high}}$ (strict inequality; ties do not count as violations), and 0 otherwise. (ii) The Spearman rank correlation ($\rho$) between severity level $\{\mathrm{low}, \mathrm{mid}, \mathrm{high}\}$ and the robustness drop. All corruptions are deterministic given fixed parameters and per-sample seeds; we do not average over multiple stochastic draws, so observed violations reflect true model behavior rather than sampling variance.

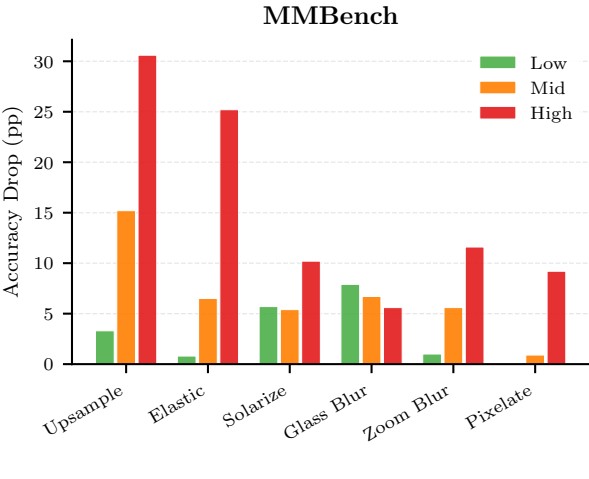

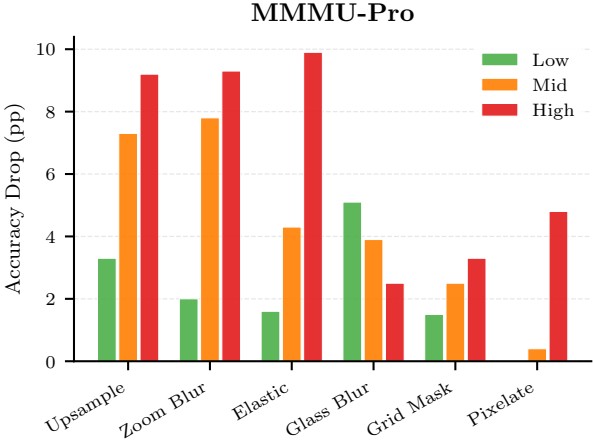

*Figure 2.* Top corruptions by severity (mean drop, 12 open-weights models). Resampling corruptions (`upsample`, `elastic_transform`) dominate at mid/high severity, while `glass_blur` shows an inverted pattern (Low > Mid > High) on both datasets.

We report these metrics in Section B.1. Overall, we observe substantial violation rates and weak-to-moderate correlations, indicating that visually ordered severity is an unreliable proxy for model difficulty, particularly on reasoning-oriented MMMU-Pro.

### 5.5. Catastrophic vs. Mild Distribution

Let $\Delta_{A_k,s} = \mathrm{Acc}_{\mathrm{clean}} - \mathrm{Acc}_{A_k,s}$ denote the accuracy drop (percentage points). We define tiers: *benign* ($\Delta \leq 1$), *mild* ($1 < \Delta \leq 3$), *moderate* ($3 < \Delta \leq 10$), *catastrophic* ($\Delta > 10$), and *positive* ($\Delta < 0$, i.e., corruption improves accuracy). Table 18 reports tier distributions by severity level (direct mode, 12 models × 42 severity-based augmentations per level, plus 12 × 7 binary). A key finding: *binary augmentations on MMBench have the highest catastrophic count (13)*, despite being trivial transforms—further evidence that spatial manipulations (flips) and color inversions

are disproportionately harmful.

### 5.6. RCE Analysis: Severity Trends and Adversarial Regimes

RCE (defined in Section 3.1) normalizes corruption impact by visual reliance, enabling comparison across models with different VG. We report the *mean RCE over all 133 configurations*: $\mathrm{mRCE} = \frac{1}{|\mathcal{C}|} \sum_{c \in \mathcal{C}} \mathrm{RCE}_c$, and additionally report severity-sliced means (Low/Mid/High over 42 configs each) and Binary (7 configs) in Table 20.

**RCE by Severity.** Table 20 reports mean RCE across models. On MMBench, RCE escalates from 1.6% (low) to 9.4% (high) to 11.2% (binary). MMMU-Pro shows higher RCE despite lower absolute drops because its smaller Visual Gain amplifies relative impact. Notably, two configurations on MMMU-Pro exceed 100% RCE (`upsample:high` and `elastic_transform:high` for Qwen3-VL-4B), indicating truly adversarial corruptions.

**Model-Specific RCE.** Table 3 includes per-model RCE and Visual Gain. On MMBench, InternVL3.5-4B has the highest RCE (7.7%), losing nearly 8% of its visual contribution on average, while Qwen3-VL-30B is most resilient (3.5%). On MMMU-Pro, Gemma-3-12B suffers 24.2% RCE—corruptions destroy nearly a quarter of its visual benefit. Most strikingly, *Qwen3-VL-4B achieves negative RCE (−6.8%)* on MMMU-Pro, meaning corruptions *improve* performance relative to clean images, confirming its minimal visual reliance on reasoning tasks.

**Worst-Case RCE by Augmentation.** Table 5 lists the top corruptions by RCE. On MMBench, `upsample:high` destroys 64.9% of visual contribution—over half the benefit of having an image. On MMMU-Pro, `elastic_transform:high` reaches 81.0% RCE, and two configurations exceed 100% (adversarial). Binary transforms (e.g., `flip_v`) achieve 22% RCE on MMBench despite being trivial operations, confirming their outsized impact relative to visual reliance.

### 5.7. Mean Corruption Error (mCE)

Following ImageNet-C (Hendrycks & Dietterich, 2019), we compute *mean Corruption Error (mCE)* to compare model robustness against a reference baseline. For each corruption type $c$, we define:

$$\mathrm{CE}_c = \frac{\sum_s E_{c,s}^{\mathrm{model}}}{\sum_s E_{c,s}^{\mathrm{ref}}} \quad (4)$$

where $E = 1 - \mathrm{Acc}$ is the error rate. For the 42 severity-based corruptions, errors are summed over $s \in \{\mathrm{low}, \mathrm{mid}, \mathrm{high}\}$; for the 7 binary corruptions, a single error term is used. The reference model is the one with the lowest baseline accuracy (analogous to AlexNet): *Gemma-3-12B*

| Dataset | Augmentation | Sev. | RCE (%) |
|---|---|---|---|
| MMBench | upsample | high | 64.9 |
| | elastic_transform | high | 53.3 |
| | upsample | mid | 32.3 |
| | zoom_blur | high | 24.6 |
| | invert | binary | 22.0 |
| | flip_v | binary | 21.7 |
| MMMU-Pro | elastic_transform | high | 81.0 |
| | upsample | high | 76.8 |
| | zoom_blur | high | 76.3 |
| | zoom_blur | mid | 64.1 |
| | upsample | mid | 60.4 |
| | glass_blur | low | 43.0 |

*Table 5.* Top corruptions by Relative Corruption Error and corresponding severity levels.

| Model | MMBench mCE | MMMU-Pro mCE |
|---|---|---|
| Qwen3-VL-235B | **50.0** | **80.2** |
| Qwen3-VL-32B | 58.6 | 84.7 |
| Qwen3-VL-30B | 62.9 | 89.0 |
| InternVL3.5-38B | 66.4 | 86.5 |
| Qwen3-VL-8B | 71.0 | 95.0 |
| Qwen3-VL-4B | 77.5 | 98.9 |
| Molmo2-8B | 78.2 | 100.0 (ref) |
| Molmo2-4B | 79.2 | 100.0 |
| InternVL3.5-8B | 81.2 | 89.1 |
| InternVL3.5-14B | 92.0 | 85.0 |
| InternVL3.5-4B | 98.3 | 93.1 |
| Gemma-3-12B | 100.0 (ref) | 101.1 |
| GPT-5.4-mini | 85.9 | 90.6 |

*Table 6.* Mean Corruption Error (mCE) following ImageNet-C methodology. Lower is better; 100% matches the reference model. GPT-5.4-mini shown separately as a proprietary model.

for MMBench (85.3%) and *Molmo2-8B* for MMMU-Pro (31.2%). Mean CE aggregates across all 49 corruption types: $\text{mCE} = \frac{1}{49} \sum_c \text{CE}_c$.

Table 6 reveals that *Qwen3-VL-235B is the most robust model on both datasets* with only 50.0% of the reference error rate on MMBench and 80.2% on MMMU-Pro. Scaling within the Qwen family shows clear mCE improvement (77.5 at 4B $\rightarrow$ 50.0% at 235B), yet worst-case drops remain above 30 pp even for the largest model. GPT-5.4-mini (mCE 85.9%) ranks below most open-weights models on MMBench, demonstrating that proprietary training does not guarantee superior robustness. Notably, mCE rankings differ from RCE rankings because mCE compares absolute error rates across models, while RCE measures each model's degradation relative to its own visual contribution. This complementary view shows that models with high baseline accuracy (Qwen family) tend to have lower mCE, while models with high visual gain but moderate baselines (InternVL3.5) show better RCE but higher mCE. In summary, we use tail-risk metrics (worst-case drop, severe-failure

rate) for deployment risk assessment, mCE for cross-model robustness ranking, and RCE to factor out language-prior reliance.

### 5.8. Flip Ground-Truth Validity

A natural concern is whether spatial transforms (especially flips) invalidate ground-truth answers for spatially-sensitive questions (e.g., "Which direction is the baby facing?"). To quantify this, we used GPT-5.4 as a judge to classify each question as inherently spatially sensitive. Only **3–6%** of questions are sensitive (47/853 MMBench, 10/324 MMMU-Pro). After excluding all sensitive questions, 90–100% of flip-induced drops are retained (full results in Section E), confirming the drops reflect model fragility, not label invalidation.

### 5.9. Qualitative Analysis of Failure Modes

Figure 3 and Figure 14 in Section F visualize all 49 augmentations applied to a representative image. A clear pattern emerges: the most damaging corruptions are those that alter *spatial structure* rather than appearance. Low-severity spatial transformations such as glass_blur, upsample, and elastic_transform often induce larger accuracy drops than visually severe photometric distortions such as noise, compression, or color shifts. This suggests that VLMs rely heavily on spatial consistency and alignment, and are disproportionately sensitive to resampling artifacts that disrupt object boundaries or relative geometry. In contrast, degradations that preserve global structure—even when visually strong—are comparatively well tolerated.

**Why spatial fragility?** We hypothesize this vulnerability stems from the patch-based architecture of Vision Transformers underlying most VLMs. When local patch structures are rearranged or distorted by effects like glass blur or elastic transformations, the pretrained patch embeddings may become misaligned with the learned representations. Resampling operations (upsample, downsample) introduce interpolation artifacts that similarly disrupt the expected patch statistics. In contrast, photometric changes (brightness, noise, compression) preserve local spatial relationships, allowing the vision encoder to maintain coherent feature extraction. This pattern is consistent across the top rankings in Figure 2, where resampling and geometry-changing corruptions repeatedly appear among the most harmful, including at low severity.

**Flip-rate evidence.** To isolate behavioral failures beyond mean drops, we measure answer-flip rates (correct on clean $\rightarrow$ incorrect under corruption). Spatial/resampling corruptions induce substantially higher flip rates than photometric corruptions on MMBench (see Appendix A.12).

**Systemic vs. Unique Failures.** Catastrophic pairs

are largely shared across models. On MMBench, the top catastrophic pairs are consistently `upsample:high`, `elastic_transform:high`, and `upsample:mid`. MMMU-Pro catastrophic pairs are rare overall; when they occur, `zoom_blur` (at mid and high severity) and `elastic_transform:high` are the most common culprits. Most catastrophic failures are systemic rather than architecture-specific (see Section A.1 for per-model breakdowns).

**Resampling Dominates Tail Risk.** Across datasets, catastrophic pairs and worst-case augmentations are dominated by resampling or geometry-changing operations (e.g., `upsample`, `elastic_transform`, `zoom_blur`), consistent with Appendix A.1. Interpolation artifacts appear to be a primary driver of failure. We quantify their contribution to catastrophic cases ($\Delta > 10$) in Appendix B.2.

**Family-Specific Vulnerabilities**

Severe-failure shares vary by model family and do not track parameter count. On MMBench, severe-failure rates range from 3.8% (Qwen3-VL-4B/30B) to 9.8% (InternVL3.5-4B). Family-specific gaps are pronounced: `shot_noise:high` drops Gemma by 12.9 points vs Qwen by 5.6, `pixelate:high` drops InternVL by 11.8 vs Qwen by 7.1, and `downsample:high` drops InternVL by 8.9 vs Qwen by 5.3. A compact family-by-augmentation matrix is provided in Appendix A.7; detailed family-level breakdowns for representative corruptions appear in Appendix A.9.

## 6. Conclusion

We presented VLM-RobustBench, a comprehensive benchmark exposing that current VLMs are *semantically strong but spatially fragile*. Our evaluation of 15 state-of-the-art VLMs, including GPT-5.4-mini and models up to 235B parameters, across 49 augmentation types reveals several counter-intuitive findings: (1) visual severity does not predict model difficulty—low-severity `glass_blur` (8–11pp drop) outperforms most high-severity corruptions; (2) trivial binary transforms can be catastrophic—vertical flipping (10.2pp) and color inversion (10.4pp) exceed most high-severity corruptions on MMBench; (3) resampling artifacts (`upsample`, `elastic_transform`) cause catastrophic failures up to 34pp across all models, and (4) our visual reliance evaluation highlights how certain benchmarks are more visually grounded (i.e., MMBench) while others heavily rely on language priors (i.e., MMMU-Pro).

**Recommendations for VLM Development.**

1. **Geometric Data Augmentation:** Training pipelines must move beyond color jitter and mixup to include heavy resampling, elastic deformations, flips, and blur augmentations during pretraining.
2. **Robustness-Aware Evaluation:** Benchmarks should report performance on spatial corruption splits (e.g., "clean vs. flipped vs. resampled") to penalize models brittle to simple geometric changes.
3. **Visual reliance:** Model providers should provide results for truly visually grounded language inputs to showcase their models' ability to perform visually grounded inferences.
4. **Family-Specific Curricula:** Different architectures exhibit distinct vulnerability fingerprints (e.g., InternVL3.5 is flip-sensitive); training should target family-specific failure modes rather than generic noise augmentation.

## Impact Statement

This work aims to improve robustness evaluation for vision-language models. We do not anticipate direct negative societal impacts from the benchmark itself; however, insights from robustness gaps can inform safer deployment and failure mitigation in real-world applications. We consider this research particularly useful for the development of foundation models for robotics that directly leverage VLMs as their backbones (e.g., (Intelligence et al., 2026; Goyal et al., 2025), inter alia). Because these embodied systems are heavily reliant on VLMs for high-level reasoning and perception, they inherently inherit the foundational weaknesses of their backbones. These vulnerabilities are often exacerbated in physical settings, where robots are routinely exposed to diverse visual perturbations and environmental corruptions—ranging from lighting shifts to sensor noise—that can compromise safety and operational reliability.

## Acknowledgements

Rohit Saxena was supported by the Engineering and Physical Sciences Research Council (EPSRC) through the AI Hub in Generative Models (grant number EP/Y028805/1). Pasquale Minervini was partially funded by ELIAI (The Edinburgh Laboratory for Integrated Artificial Intelligence), EPSRC (grant no. EP/W002876/1), and a donation from Accenture LLP.

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

| Model | MMBench worst (aug, level, drop) | MMMU-Pro worst (aug, level, drop) |
|---|---|---|
| Qwen3-VL-4B | upsample (high, 26.3) | elastic_transform (high, 7.4) |
| Qwen3-VL-8B | upsample (high, 30.2) | elastic_transform (high, 8.9) |
| Qwen3-VL-30B | upsample (high, 29.4) | elastic_transform (high, 13.9) |
| Qwen3-VL-32B | upsample (high, 32.4) | elastic_transform (high, 13.0) |
| Qwen3-VL-235B | upsample (high, 30.7) | upsample (high, 20.7) |
| InternVL3.5-4B | upsample (high, 30.6) | zoom_blur (high, 11.1) |
| InternVL3.5-8B | upsample (high, 31.6) | zoom_blur (high, 12.3) |
| InternVL3.5-14B | upsample (high, 29.4) | zoom_blur (high, 9.6) |
| InternVL3.5-38B | upsample (high, 27.2) | zoom_blur (high, 9.6) |
| Molmo2-4B | upsample (high, 33.1) | upsample (high, 5.6) |
| Molmo2-8B | upsample (high, 33.9) | upsample (high, 5.6) |
| Gemma-3-12B | upsample (high, 32.1) | upsample (high, 9.9) |
| GPT-5.4-mini | upsample (high, 31.8) | elastic_transform (high, 13.0) |
| Qwen3-VL-4B-Thinking | upsample (high, 29.5) | upsample (mid, 19.1) |
| Qwen3-VL-8B-Thinking | upsample (high, 30.8) | upsample (high, 23.1) |

*Table 7.* Worst-case augmentation per model. Drop is baseline minus accuracy under that augmentation (percentage points); "bin" denotes binary augmentations.

| MMBench | | MMMU-Pro | |
|---|---|---|---|
| **Category** | **Drop** | **Subject** | **Drop** |
| image_style | 5.30 | Art | 4.75 |
| attribute_comparison | 5.26 | Music | 3.90 |
| structuralized_imagetext_understanding | 4.81 | Economics | 3.69 |
| social_relation | 4.57 | Art_Theory | 3.46 |
| nature_relation | 4.51 | Pharmacy | 3.35 |

*Table 8.* Top-5 dataset categories with the largest average drops (percentage points), aggregated across models in direct mode.

# A. Additional Results

### A.1. Worst-Case Augmentations

### A.2. Dataset Category Sensitivity

### A.3. Scaling Trends

### A.4. Prompting-Mode Performance (Qwen)

### A.5. Prompting-Mode Tier Distributions (Qwen)

### A.6. Positive Augmentations

A small number of augmentations yield negative $\Delta$ (i.e., higher accuracy than baseline). On MMBench, `brightness:low/mid`, `gamma_up:low/mid`, and `gaussian_noise:low` show marginal gains ($-0.1$ to $-0.2$ pp); on MMMU-Pro, `hue_shift:low`, `gaussian_blur:low`, and `speckle_noise:low/mid` exhibit similar small effects. Given the magnitude ($<0.5$ pp), these may reflect noise, mild regularization, or dataset-specific priors rather than robust improvements; we note them for completeness but do not draw strong conclusions.

### A.7. Family-Level Vulnerability Matrix

### A.8. Per-Model Top-5 Corruptions

Tables 13 and 14 provide per-model top-5 most harmful corruptions at each severity. The aggregate patterns from Figure 2 hold consistently across models: `glass_blur` dominates at low severity, while `upsample` and `elastic_transform` dominate at mid/high severities.

| Family | MMMU-Pro slope ($R^2$, n) | MMBench slope ($R^2$, n) |
|---|---|---|
| Qwen3-VL | +2.95 (1.00, n=3) | -0.38 (0.17, n=3) |
| InternVL3.5 | -0.94 (0.12, n=3) | -1.44 (0.89, n=3) |
| Molmo2 | -1.66 (1.00, n=2) | -1.00 (1.00, n=2) |

*Table 9.* Scaling of robustness drop with model size within families (direct mode). Slope is the change in drop per log10(parameters); negative values indicate improved robustness with scale.

| Model | MMMU-Pro | | MMBench | |
|---|---|---|---|---|
| | Baseline | Drop | Baseline | Drop |
| Qwen3-VL-4B-COT | 32.1 | +1.3 | 86.9 | +2.5 |
| Qwen3-VL-8B-COT | 42.0 | +3.1 | 89.6 | +3.2 |
| Qwen3-VL-4B-Thinking | 43.5 | +2.5 | 89.3 | +1.8 |
| Qwen3-VL-8B-Thinking | 50.0 | +3.8 | 91.1 | +3.0 |

*Table 10.* Baseline accuracy and mean drop for Qwen models under COT and thinking modes. Drop is averaged over the 133 corrupted configurations. Direct-mode results are in Table 3.

*Table 13.* Per-model top-5 most harmful corruptions at each severity (MMBench, direct mode). GPT-5.4-mini shown below the rule as a proprietary baseline.

| Model | Low | | Mid | | High | |
|---|---|---|---|---|---|---|
| Gemma-3-12B | glass_blur | (10.7), | upsample | (15.2), | upsample | (32.1), |
| | solarize | (3.6), | shot_noise | (8.4), | elastic_transform | (23.4), |
| | shot_noise | (2.5), | glass_blur | (8.2), | zoom_blur | (13.2), |
| | upsample | (2.5), | elastic_transform | (6.7), | shot_noise | (12.9), |
| | text_overlay (2.3) | | zoom_blur (6.0) | | pixelate (11.4) | |
| InternVL3.5-4B | glass_blur | (10.7), | upsample | (17.6), | upsample | (30.6), |
| | solarize | (7.3), | glass_blur | (9.6), | elastic_transform | (25.3), |
| | upsample | (3.9), | elastic_transform | (8.6), | solarize | (14.9), |
| | shot_noise | (3.2), | solarize | (7.3), | pixelate | (13.0), |
| | zoom_blur (2.3) | | zoom_blur (7.3) | | shot_noise (12.7) | |
| InternVL3.5-8B | glass_blur | (9.3), | upsample | (17.1), | upsample | (31.6), |
| | solarize | (8.1), | elastic_transform | (8.7), | elastic_transform | (28.8), |
| | upsample | (4.5), | zoom_blur | (8.0), | zoom_blur | (15.9), |
| | shot_noise | (3.4), | glass_blur | (7.4), | shot_noise | (13.7), |
| | grid_mask (2.8) | | solarize (7.3) | | pixelate (12.7) | |
| InternVL3.5-14B | glass_blur | (9.5), | upsample | (17.9), | upsample | (29.4), |
| | solarize | (5.3), | glass_blur | (7.8), | elastic_transform | (24.7), |
| | shot_noise | (3.3), | elastic_transform | (6.6), | pixelate | (13.2), |
| | upsample | (3.2), | zoom_blur | (6.5), | zoom_blur | (12.5), |
| | grid_mask (1.1) | | solarize (5.5) | | solarize (11.4) | |
| Molmo2-4B | glass_blur | (7.4), | upsample | (15.9), | upsample | (33.1), |
| | solarize | (5.5), | zoom_blur | (6.3), | elastic_transform | (23.9), |
| | upsample | (4.1), | glass_blur | (5.7), | zoom_blur | (10.0), |
| | shot_noise | (2.1), | elastic_transform | (4.9), | center_occlusion | (9.0), |
| | grid_mask (1.9) | | solarize (4.3) | | pixelate (7.8) | |
| Molmo2-8B | glass_blur | (6.3), | upsample | (18.9), | upsample | (33.9), |
| | solarize | (4.6), | glass_blur | (6.7), | elastic_transform | (25.6), |
| | upsample | (3.5), | elastic_transform | (6.6), | zoom_blur | (9.4), |
| | grid_mask | (1.1), | zoom_blur | (4.5), | solarize (9.1), pixelate (8.7) | |
| | zoom_blur (0.6) | | solarize (4.3) | | | |
| Qwen3-VL-4B | glass_blur | (7.0), | upsample | (13.1), | upsample | (26.3), |
| | solarize | (4.5), | elastic_transform | (5.7), | elastic_transform | (25.8), |
| | upsample | (2.6), | glass_blur | (5.4), | zoom_blur | (9.6), |
| | random_occlusion | (1.2), | zoom_blur | (4.5), | solarize | (8.3), |
| | elastic_transform (1.1) | | solarize (4.1) | | center_occlusion (6.0) | |

*Table 13 continued from previous page*

| Model | Low | Mid | High |
|---|---|---|---|
| Qwen3-VL-8B | glass_blur (8.2), solarize (5.9), grid_mask (2.3), shot_noise (1.8), upsample (1.8) | upsample (14.2), glass_blur (7.0), elastic_transform (6.1), solarize (6.0), zoom_blur (5.2) | upsample (30.2), elastic_transform (27.0), zoom_blur (12.3), solarize (10.3), pixelate (7.8) |
| Qwen3-VL-30B | solarize (6.1), glass_blur (5.6), upsample (2.6), random_occlusion (1.2), watermark (1.1) | upsample (11.8), elastic_transform (6.2), solarize (4.9), glass_blur (4.3), zoom_blur (4.3) | upsample (29.4), elastic_transform (23.2), zoom_blur (10.9), solarize (8.4), center_occlusion (5.9) |
| Qwen3-VL-32B | glass_blur (8.1), solarize (6.9), upsample (3.8), grid_mask (1.9), shot_noise (1.4) | upsample (14.2), glass_blur (7.6), elastic_transform (7.2), solarize (6.0), zoom_blur (5.7) | upsample (32.4), elastic_transform (26.6), zoom_blur (14.2), solarize (10.7), pixelate (9.7) |
| Qwen3-VL-235B | glass_blur (6.3), solarize (6.2), upsample (4.6), grid_mask (1.9), shot_noise (1.5) | upsample (14.5), elastic_transform (6.1), glass_blur (5.4), solarize (4.9), zoom_blur (4.9) | upsample (30.7), elastic_transform (26.6), solarize (10.1), zoom_blur (10.1), pixelate (7.4) |
| InternVL3.5-38B | glass_blur (6.1), solarize (4.6), upsample (2.2), shot_noise (1.8), grid_mask (1.4) | upsample (11.7), elastic_transform (5.2), glass_blur (4.8), solarize (4.8), zoom_blur (4.3) | upsample (27.2), elastic_transform (21.0), zoom_blur (9.5), pixelate (8.3), solarize (8.2) |
| GPT-5.4-mini | glass_blur (11.7), solarize (6.9), shot_noise (4.8), snow (2.9), upsample (2.5) | upsample (14.8), glass_blur (10.8), elastic_transform (8.2), shot_noise (6.8), solarize (6.5) | upsample (31.8), elastic_transform (25.3), shot_noise (12.4), solarize (11.1), zoom_blur (10.1) |

*Table 14.* Per-model top-5 most harmful corruptions at each severity (MMMU-Pro, direct mode). GPT-5.4-mini shown below the rule as a proprietary baseline.

| Model | Low | Mid | High |
|---|---|---|---|
| Gemma-3-12B | glass_blur (8.9), solarize (4.9), brightness (3.7), grid_mask (3.7), defocus_blur (3.4) | upsample (9.6), zoom_blur (8.9), elastic_transform (5.2), glass_blur (4.6), motion_blur (4.6) | upsample (9.9), zoom_blur (9.9), elastic_transform (9.3), downsample (5.9), center_occlusion (5.6) |
| InternVL3.5-4B | glass_blur (5.6), upsample (5.6), grid_mask (3.7), solarize (3.1), watermark (2.5) | zoom_blur (9.6), upsample (6.5), glass_blur (5.2), grid_mask (4.9), elastic_transform (4.6) | zoom_blur (11.1), elastic_transform (9.3), upsample (8.9), motion_blur (5.9), downsample (5.6) |
| InternVL3.5-8B | glass_blur (5.6), zoom_blur (4.6), grid_mask (4.3), motion_blur (3.7), rotate (2.8) | zoom_blur (11.7), upsample (8.0), random_occlusion (4.9), glass_blur (4.6), motion_blur (4.6) | zoom_blur (12.3), elastic_transform (10.5), upsample (9.6), pixelate (7.7), downsample (6.5) |
| InternVL3.5-14B | glass_blur (5.9), upsample (3.4), snow (1.5), zoom_blur (1.5), grid_mask (1.2) | zoom_blur (8.3), upsample (7.4), glass_blur (5.2), rotate (2.8), grid_mask (2.5) | zoom_blur (9.6), elastic_transform (9.3), upsample (9.3), pixelate (5.6), downsample (4.6) |

*Table 14 continued from previous page*

| Model | Low | Mid | High |
|---|---|---|---|
| Molmo2-4B | glass_blur (4.3), rotate (1.5), shot_noise (1.2), add_border (0.9), brightness_up (0.9) | upsample (4.3), glass_blur (3.1), zoom_blur (3.1), brightness_up (1.9), saturation (1.9) | upsample (5.6), zoom_blur (4.6), elastic_transform (4.0), downsample (2.5), brightness (1.9) |
| Molmo2-8B | glass_blur (4.0), zoom_blur (1.9), upsample (1.5), center_occlusion (1.2), gamma_up (0.6) | upsample (3.7), zoom_blur (3.7), elastic_transform (2.5), rotate (2.5), glass_blur (2.2) | upsample (5.6), elastic_transform (4.9), zoom_blur (4.6), rotate (3.4), gamma_up (1.9) |
| Qwen3-VL-4B | glass_blur (2.8), upsample (1.5), gamma (0.6), saturation (0.3), sharpen (0.0) | upsample (5.2), zoom_blur (5.2), glass_blur (2.2), solarize (1.5), downsample (0.9) | elastic_transform (7.4), upsample (7.4), zoom_blur (6.5), downsample (2.2), salt_pepper (2.2) |
| Qwen3-VL-8B | glass_blur (5.2), watermark (2.8), upsample (2.2), salt_pepper (1.2), snow (1.2) | zoom_blur (8.0), upsample (7.4), watermark (4.0), glass_blur (3.7), elastic_transform (2.2) | elastic_transform (8.9), upsample (8.9), zoom_blur (8.6), pixelate (5.2), brightness_up (2.5) |
| Qwen3-VL-30B | glass_blur (6.8), upsample (6.2), grid_mask (3.7), affine (2.8), watermark (2.8) | upsample (12.0), zoom_blur (12.0), elastic_transform (6.2), glass_blur (5.2), grid_mask (4.9) | elastic_transform (13.9), zoom_blur (13.9), upsample (12.7), grid_mask (7.4), pixelate (6.5) |
| Qwen3-VL-32B | glass_blur (4.9), solarize (3.7), upsample (3.7), zoom_blur (3.1), affine (1.9) | zoom_blur (10.2), upsample (8.6), elastic_transform (7.1), grid_mask (3.4), glass_blur (3.1) | upsample (13.0), elastic_transform (12.3), zoom_blur (10.2), pixelate (4.9), downsample (4.3) |
| Qwen3-VL-235B | elastic_transform (13.6), jpeg_compression (13.3), spatter (12.3), upsample (5.6), glass_blur (4.3) | elastic_transform (15.1), spatter (12.3), jpeg_compression (11.7), upsample (8.0), zoom_blur (7.7) | elastic_transform (20.7), jpeg_compression (13.0), spatter (12.7), upsample (11.1), zoom_blur (10.8) |
| InternVL3.5-38B | upsample (4.0), glass_blur (3.1), zoom_blur (3.1), watermark (1.9), gamma (1.2) | upsample (7.1), zoom_blur (5.6), elastic_transform (4.0), glass_blur (3.1), snow (2.2) | zoom_blur (9.6), upsample (8.6), elastic_transform (8.0), pixelate (6.2), grid_mask (3.7) |
| GPT-5.4-mini | glass_blur (9.6), perspective_transform (3.4), solarize (3.1), upsample (3.1), random_occlusion (2.5) | upsample (8.9), zoom_blur (7.7), glass_blur (7.4), elastic_transform (6.5), motion_blur (3.7) | elastic_transform (13.0), zoom_blur (10.8), upsample (9.9), downsample (8.6), glass_blur (8.0) |

## A.9. Detailed Robustness Results by Family

We provide the complete breakdown of accuracy drops for key augmentation types across the four model families. Values represent the family-averaged drop (percentage points) at Low, Mid, and High severity on MMBench.

## A.10. Tier Distributions

Table 18 reports tier distributions by severity level (direct mode, 12 models × 42 severity-based augmentations per level, plus 12 × 7 binary).

Table 19 breaks down tier shares per model, highlighting that catastrophic and positive rates vary widely even within families.

| Mode | Mild | Moderate | Catastrophic | Positive |
|------|------|----------|--------------|----------|
| **MMBench (Qwen models)** | | | | |
| Direct | 23.6 | 17.0 | 3.5 | 21.6 |
| COT | 28.9 | 23.3 | 5.6 | 16.9 |
| Thinking | 27.1 | 21.1 | 4.1 | 16.2 |
| **MMMU-Pro (Qwen models)** | | | | |
| Direct | 19.8 | 10.3 | 1.3 | 42.1 |
| COT | 19.5 | 21.4 | 6.4 | 26.7 |
| Thinking | 18.8 | 19.9 | 13.2 | 27.8 |

*Table 11.* Tier shares (%) for Qwen models by prompting mode.

| Aug-Sev | Gemma | Qwen | InternVL | Molmo | GPT-5.4 |
|---------|-------|------|----------|-------|---------|
| shot_noise:high | 12.90 | 5.58 | 11.20 | 5.62 | 12.43 |
| pixelate:high | 11.37 | 7.06 | 11.81 | 8.26 | 9.26 |
| downsample:high | 8.79 | 5.25 | 8.85 | 6.74 | 8.44 |
| solarize:high | 10.08 | 9.57 | 11.75 | 8.50 | 11.14 |

*Table 12.* Family-level mean drops (MMBench, direct) for selected aug-level pairs.

## A.11. RCE by Severity

Table 20 reports mean RCE across models by severity level. Higher RCE on MMMU-Pro reflects its smaller Visual Gain denominator. Two configurations on MMMU-Pro exceed 100% RCE (`upsample:high` and `elastic_transform:high` for Qwen3-VL-4B), indicating truly adversarial corruptions.

## A.12. Per-Example Flip Decomposition

Table 21 shows answer-flip rates (fraction of questions correct on clean that become incorrect under corruption) for a representative model (Qwen3-VL-8B) on MMBench. Spatial/resampling corruptions cause substantially more flips than photometric ones, even when the latter are at high severity.

Accuracy drop $\Delta$ conflates two opposing effects: examples the model previously answered correctly but now fails (*harmful flips*, $\text{Flip}^+$), and examples it previously failed but now succeeds (*helpful flips*, $\text{Flip}^-$):

$$\text{Flip}^+ = \text{Pr}(\text{correct}_{\text{clean}} \wedge \text{wrong}_{\text{corrupted}}) \tag{5}$$

$$\text{Flip}^- = \text{Pr}(\text{wrong}_{\text{clean}} \wedge \text{correct}_{\text{corrupted}}) \tag{6}$$

with net accuracy drop $\Delta = \text{Flip}^+ - \text{Flip}^-$. Flip rates are computed per example then averaged across the dataset; when aggregating across models or corruptions, we report macro-averages. This decomposition reveals whether drops stem from genuine failures or are partially masked by compensating gains.

**Flip Rates by Severity.** Table 22 reports flip rates aggregated by severity level. On MMBench, harmful flips escalate sharply (1.7% at low → 6.1% at high), while helpful flips remain low (1.0–1.7%), confirming that accuracy drops reflect genuine failures rather than noise. Binary augmentations show the highest harmful flip rate (6.7%) with minimal compensation (1.4% $\text{Flip}^-$). For severity-based corruptions, MMMU-Pro exhibits smaller $\text{Flip}^+/\text{Flip}^-$ ratios (1.3–1.7× vs. 1.8–3.6× on MMBench), consistent with its lower visual reliance.

**Model-Specific Patterns.** Table 23 reveals striking model differences. On MMBench, Molmo2-8B exhibits the highest $\text{Flip}^+/\text{Flip}^-$ ratio (3.81), indicating "clean" degradation with minimal lucky compensations. Gemma-3-12B has the highest absolute $\text{Flip}^+$ rate (5.21%), making it the most fragile. On MMMU-Pro, Gemma-3-12B again leads in $\text{Flip}^+$ (5.09%), while Qwen3-VL-4B achieves a ratio below 1.0 (0.83), meaning corruptions *help more than hurt*—strong evidence of minimal visual reliance on reasoning tasks.

**Binary Augmentation Flips.** Table 24 drills into per-augmentation flip rates for binary transforms. On MMBench, `flip_v` and `invert` consistently cause 10–15% harmful flips across models, with InternVL3.5-4B reaching 15.7% for

*Table 15.* **Qwen3-VL Family:** Mean accuracy drops on MMBench. Note the resilience to noise (e.g., Gaussian Noise) vs. fragility to resampling (Upsample).

| Augmentation | Low | Mid | High |
|---|---|---|---|
| Upsample | 2.31 | 13.05 | 28.65 |
| Elastic Transform | 0.78 | 6.02 | 25.32 |
| Zoom Blur | 0.55 | 4.65 | 10.94 |
| Solarize | 5.47 | 5.00 | 9.03 |
| Glass Blur | 6.96 | 5.59 | 4.10 |
| Pixelate | 0.16 | 0.35 | 6.06 |
| Shot Noise | 0.55 | 2.19 | 5.12 |
| Brightness | 0.23 | 0.23 | 3.99 |
| JPEG Compression | 0.20 | 0.27 | 0.31 |

*Table 16.* **InternVL3.5 Family:** Mean accuracy drops on MMBench. This family shows higher sensitivity to pixelation and noise compared to Qwen.

| Augmentation | Low | Mid | High |
|---|---|---|---|
| Upsample | 3.83 | 17.55 | 30.56 |
| Elastic Transform | 0.78 | 7.94 | 26.30 |
| Zoom Blur | 1.60 | 7.23 | 13.36 |
| Pixelate | 0.55 | 1.60 | 12.97 |
| Solarize | 6.88 | 6.68 | 12.93 |
| Shot Noise | 3.28 | 5.98 | 12.39 |
| Glass Blur | 9.81 | 8.28 | 7.11 |
| Motion Blur | 0.74 | 3.05 | 7.31 |
| JPEG Compression | 0.20 | 0.27 | 0.63 |

vertical flip. On MMMU-Pro, the same augmentations show dramatically lower $\text{Flip}^+$ (5–11%) and higher $\text{Flip}^-$, with some models (Molmo2-8B) showing near-zero or negative net flips for `flip_h`—confirming that spatial transforms harm perception far more than reasoning.

## A.13. Answer-Flip Rates Across Models

Table 25 extends the flip-rate analysis to all direct-mode models for selected corruptions. Flip rate is defined as the fraction of questions answered correctly on clean images that become incorrect under corruption.

*Table 17.* **Gemma 3 & Molmo 2 Families:** Comparison of key failure modes (High Severity Drops).

| Augmentation (High) | Gemma 3 (12B) | Molmo 2 (Avg) |
|---|---|---|
| Upsample | 32.12 | 33.47 |
| Elastic Transform | 23.45 | 24.74 |
| Zoom Blur | 13.25 | 9.67 |
| Shot Noise | 12.90 | 5.62 |
| Pixelate | 11.37 | 8.26 |
| Solarize | 10.08 | 8.50 |
| Glass Blur | 5.51 | 6.10 |

| Dataset | Severity | Benign | Mild | Moderate | Catastrophic | Positive |
|---|---|---|---|---|---|---|
| MMBench | Low | 296 | 65 | 32 | 2 | 109 |
| | Mid | 188 | 157 | 81 | 12 | 66 |
| | High | 105 | 151 | 178 | 48 | 22 |
| | Binary | 8 | 18 | 41 | **13** | 4 |
| MMMU-Pro | Low | 182 | 91 | 34 | 3 | 194 |
| | Mid | 158 | 119 | 63 | 7 | 157 |
| | High | 137 | 135 | 99 | 14 | 119 |
| | Binary | 24 | 18 | 16 | 0 | 26 |

*Table 18.* Tier distribution by severity level (counts out of 504 for severity-based, 84 for binary; rows sum to the total). Tiers use fixed thresholds: Catastrophic = $\Delta > 10$pp; Positive = $\Delta < 0$ (corruption improves accuracy). Binary augmentations on MMBench produce 13 catastrophic cases—exceeding low-severity—driven by vertical flip and color inversion. MMMU-Pro's high Positive count at low severity (194/504) reflects greater reliance on language priors.

| MMBench (Direct) | | | | |
|---|---|---|---|---|
| **Model** | **Mild %** | **Moderate %** | **Catastrophic Rate (%)** | **Positive %** |
| Qwen3-VL-4B | 21.8 | 18.0 | 2.3 | 24.1 |
| Qwen3-VL-8B | 30.8 | 19.5 | 4.5 | 11.3 |
| Qwen3-VL-30B | 18.0 | 13.5 | 3.8 | 29.3 |
| Qwen3-VL-32B | 24.8 | 21.1 | 5.3 | 6.0 |
| Qwen3-VL-235B | 18.8 | 21.8 | 4.5 | 15.8 |
| InternVL3.5-4B | 29.3 | 30.1 | 8.3 | 3.8 |
| InternVL3.5-8B | 26.3 | 28.6 | 6.8 | 11.3 |
| InternVL3.5-14B | 18.0 | 21.1 | 6.8 | 14.3 |
| InternVL3.5-38B | 22.6 | 18.8 | 3.0 | 3.8 |
| Molmo2-4B | 27.1 | 21.1 | 2.3 | 8.3 |
| Molmo2-8B | 21.1 | 17.3 | 2.3 | 17.3 |
| Gemma-3-12B | 35.3 | 18.8 | 6.8 | 6.0 |
| GPT-5.4-mini | 21.1 | 23.3 | 6.8 | 23.3 |

| MMMU-Pro (Direct) | | | | |
|---|---|---|---|---|
| **Model** | **Mild %** | **Moderate %** | **Catastrophic Rate (%)** | **Positive %** |
| Qwen3-VL-4B | 9.0 | 3.8 | 0.0 | 74.4 |
| Qwen3-VL-8B | 15.0 | 7.5 | 0.0 | 39.8 |
| Qwen3-VL-30B | 35.3 | 19.5 | 3.8 | 12.0 |
| Qwen3-VL-32B | 27.1 | 15.8 | 3.0 | 21.1 |
| Qwen3-VL-235B | 10.5 | 6.8 | 8.3 | 52.6 |
| InternVL3.5-4B | 24.8 | 16.5 | 0.8 | 26.3 |
| InternVL3.5-8B | 38.3 | 24.8 | 2.3 | 6.0 |
| InternVL3.5-14B | 17.3 | 10.5 | 0.0 | 37.6 |
| InternVL3.5-38B | 17.3 | 11.3 | 0.0 | 29.3 |
| Molmo2-4B | 15.8 | 5.3 | 0.0 | 19.5 |
| Molmo2-8B | 12.8 | 5.3 | 0.0 | 51.1 |
| Gemma-3-12B | 49.6 | 32.3 | 0.0 | 3.0 |
| GPT-5.4-mini | 25.6 | 21.8 | 1.5 | 24.8 |

*Table 19.* Per-model tier shares (%) across 133 augmentation configurations (direct mode). Tiers use fixed thresholds: Catastrophic = $\Delta > 10$pp (distinct from the relative Severe-Failure Rate in Table 3). Benign shares are omitted for brevity.

| Dataset | Severity | Mean RCE (%) | Interpretation |
|---|---|---|---|
| MMBench | Low | 1.6 | Minimal visual loss |
| | Mid | 3.9 | Moderate impact |
| | High | 9.4 | ~10% visual loss |
| | Binary | 11.2 | Highest relative harm |
| MMMU-Pro | Low | 3.6 | Low but > MMBench |
| | Mid | 7.4 | Moderate |
| | High | 14.0 | Severe relative loss |
| | Binary | 10.5 | High relative harm |

*Table 20.* Mean Relative Corruption Error by severity. RCE measures what fraction of visual contribution is destroyed. Higher RCE on MMMU-Pro reflects its smaller Visual Gain denominator.

| Corruption | Severity | Flip rate |
|---|---|---|
| glass_blur | low | 11.7% |
| upsample | high | 36.3% |
| elastic_transform | high | 32.9% |
| brightness | high | 4.4% |
| jpeg_compression | high | 1.6% |

*Table 21.* Answer-flip rates for Qwen3-VL-8B on MMBench. Spatial/resampling corruptions (top) cause substantially more flips than photometric ones (bottom), even at high severity.

| Dataset | Severity | Flip$^+$ (%) | Flip$^-$ (%) | Ratio |
|---|---|---|---|---|
| MMBench | Low | 1.71 | 0.97 | 1.77 |
| | Mid | 3.23 | 1.39 | 2.32 |
| | High | 6.11 | 1.68 | 3.63 |
| | Binary | 6.66 | 1.36 | 4.91 |
| MMMU-Pro | Low | 2.19 | 1.65 | 1.32 |
| | Mid | 3.22 | 2.19 | 1.47 |
| | High | 4.49 | 2.68 | 1.67 |
| | Binary | 3.58 | 2.18 | 1.64 |

*Table 22.* Flip rates by severity level (averaged over models). Flip$^+$ = harmful (correct→wrong), Flip$^-$ = helpful (wrong→correct). Higher ratios indicate "purer" degradation with less compensating gains.

| | MMBench | | MMMU-Pro | |
|---|---|---|---|---|
| Model | Flip$^+$ | Ratio | Flip$^+$ | Ratio |
| Qwen3-VL-4B | 3.33 | 2.32 | 2.29 | **0.83** |
| Qwen3-VL-8B | 4.15 | 2.59 | 3.54 | 1.18 |
| Qwen3-VL-30B | 3.36 | 2.00 | 4.03 | 2.03 |
| Qwen3-VL-32B | 3.85 | 3.24 | 3.91 | 1.60 |
| Qwen3-VL-235B | 3.30 | 3.11 | 3.69 | 1.53 |
| InternVL3.5-4B | 4.81 | 3.58 | 3.31 | 1.76 |
| InternVL3.5-8B | 4.91 | 3.05 | 3.68 | 2.71 |
| InternVL3.5-14B | 4.12 | 2.82 | 2.81 | 1.39 |
| InternVL3.5-38B | 2.86 | 3.75 | 2.35 | 1.59 |
| Molmo2-4B | 3.31 | 3.65 | 2.65 | 1.29 |
| Molmo2-8B | 2.88 | **3.81** | 2.43 | 1.03 |
| Gemma-3-12B | **5.21** | 2.19 | **5.09** | 2.14 |

*Table 23.* Model flip rates (%) and Flip$^+$/Flip$^-$ ratios. Higher ratios indicate purer degradation. Bold: highest Flip$^+$ (most fragile) and extreme ratios.

| | MMBench | | | MMMU-Pro | | |
|---|---|---|---|---|---|---|
| Augmentation | Flip$^+$ | Flip$^-$ | Net | Flip$^+$ | Flip$^-$ | Net |
| flip_v | 12.2 | 1.9 | 10.2 | 8.5 | 3.8 | 4.8 |
| flip_h | 8.7 | 1.7 | 7.0 | 7.1 | 3.5 | 3.6 |
| invert | 12.0 | 1.6 | 10.4 | 3.7 | 2.4 | 1.3 |
| channel_swap | 4.2 | 1.2 | 3.0 | 1.1 | 1.1 | −0.1 |
| equalize | 4.7 | 1.5 | 3.1 | 2.8 | 2.4 | 0.4 |
| grayscale | 4.6 | 1.4 | 3.2 | 1.6 | 1.5 | 0.1 |
| autocontrast | 0.2 | 0.2 | 0.0 | 0.3 | 0.6 | −0.2 |

*Table 24.* Binary augmentation flip rates (%) averaged over 12 open-weights models. Vertical flip and invert dominate harmful flips on MMBench but show reduced impact on MMMU-Pro.

| Model | glass:low | ups:high | elast:high | bright:high | jpeg:high |
|---|---|---|---|---|---|
| Qwen3-VL-4B | 8.4 | 29.2 | 29.1 | 2.7 | 1.4 |
| Qwen3-VL-8B | 10.6 | 32.7 | 29.7 | 4.0 | 1.4 |
| Qwen3-VL-30B | 8.6 | 31.2 | 26.7 | 2.5 | 1.5 |
| Qwen3-VL-32B | 10.0 | 34.5 | 28.5 | 3.6 | 1.6 |
| Qwen3-VL-235B | 8.1 | 32.0 | 29.0 | 2.2 | 0.9 |
| InternVL3.5-4B | 12.8 | 33.2 | 27.9 | 4.8 | 2.3 |
| InternVL3.5-8B | 12.4 | 34.7 | 32.0 | 5.5 | 1.9 |
| InternVL3.5-14B | 12.2 | 32.4 | 27.2 | 3.9 | 1.4 |
| InternVL3.5-38B | 8.0 | 29.1 | 23.4 | 3.4 | 0.8 |
| Molmo2-4B | 9.0 | 34.7 | 26.4 | 3.9 | 0.8 |
| Molmo2-8B | 7.9 | 35.4 | 27.1 | 3.8 | 1.1 |
| Gemma-3-12B | 14.2 | 35.8 | 26.6 | 6.6 | 4.3 |

*Table 25.* Answer-flip rates (%) across 12 open-weights models on MMBench. Spatial/resampling corruptions (columns 2–4) consistently cause higher flip rates than photometric ones (columns 5–6).

## A.14. GPT-5.4-mini: Proprietary Baseline

We include GPT-5.4-mini as a proprietary baseline, evaluated identically to the open-weights models. Its results are shown below the midrule in per-model tables throughout the paper but excluded from all averaged analyses to avoid conflating open-weights and proprietary results.

**Key findings.** GPT-5.4-mini achieves moderate baseline accuracy (87.7% MMBench, 39.2% MMMU-Pro) but exhibits notable robustness weaknesses:

- **Highest Worst@Low** (11.7pp on MMBench)—the most fragile model at low severity, driven by `glass_blur` (11.7pp drop).
- **Highest Flip$^+$ rate** among all instruct models: 5.23% on MMBench and 6.46% on MMMU-Pro, indicating the most answer instability under corruption.
- **mCE of 85.9%** on MMBench, ranking below most open-weights models and above only InternVL3.5-14B (92.0%) and InternVL3.5-4B (98.3%).
- Family-level vulnerability pattern similar to Gemma-3-12B: high sensitivity to `shot_noise` (12.4pp), `solarize` (11.1pp), and `pixelate` (9.3pp).

These results demonstrate that proprietary training does not guarantee superior robustness. GPT-5.4-mini's high flip rate and worst-case low-severity vulnerability suggest that its training may prioritize clean-image performance without explicit robustness considerations.

## B. Quantitative Robustness Metrics

### B.1. Severity Mismatch Metrics

Table 26 reports the consistency between visual severity levels and model performance drops. A high violation rate indicates that increasing visual severity does not reliably lead to larger performance drops.

*Table 26.* Severity mismatch metrics. **Violation Rate**: Fraction of augmentation trajectories where drop does not strictly increase with severity. **Mean Spearman** $\rho$: Rank correlation between severity and drop (averaged across models/augmentations).

| Dataset | Violation Rate (%) | Mean Spearman $\rho$ |
|---|---|---|
| MMBench | 30.2 | 0.71 |
| MMMU-Pro | 56.1 | 0.34 |

### B.2. Tail Risk Share from Spatial/Resampling Corruptions

Table 27 quantifies the contribution of spatial/resampling corruptions to catastrophic failures. We define "Spatial/Resampling" augmentations as: `upsample`, `downsample`, `elastic_transform`, `zoom_blur`, `rotate`, `shear`, `affine`, `perspective_transform`, and `pixelate` (included as a resolution/resampling artifact that disrupts spatial structure).

*Table 27.* Fraction of catastrophic cases ($\Delta > 10$) attributable to spatial/resampling corruptions.

| Dataset | Share from Spatial/Resampling (%) | Top Contributors |
|---|---|---|
| MMBench | 65.5 | `upsample`, `elastic_transform`, `zoom_blur` |
| MMMU-Pro | 100.0 | `zoom_blur`, `elastic_transform`, `upsample` |

### B.3. Mean Corruption Error by Category

Following ImageNet-C methodology, we compute mean Corruption Error (mCE) to compare model robustness against a reference baseline. For each corruption type $c$, $\text{CE}_c = \frac{\sum_s E_{c,s}^{\text{model}}}{\sum_s E_{c,s}^{\text{ref}}}$ where $E = 1 - \text{Acc}$ is the error rate. The reference model is the one with lowest baseline accuracy: **Gemma-3-12B** (85.3%) for MMBench and **Molmo2-8B** (31.2%) for MMMU-Pro. Values below 100% indicate better robustness than the reference; values above 100% indicate worse robustness.

Table 28 reveals category-specific robustness patterns. On MMBench, **Qwen3-VL-235B** achieves the lowest mCE across all categories (43–58%), demonstrating consistent robustness. Noise corruptions show the best relative robustness, while Binary

transforms show the worst. Notably, InternVL3.5-4B approaches or exceeds 100% mCE in most categories (Blur 101.1%, Binary 100.8%), indicating it is less robust than the reference Gemma model despite having similar baseline accuracy.

On MMMU-Pro, all models cluster near 100% mCE (range 80–102%), reflecting the harder dataset where even the reference model struggles. **Qwen3-VL-235B** leads with the lowest overall mCE (80.2%), followed by Qwen3-VL-32B (84.7%) and InternVL3.5-14B (85.0%). Conversely, Gemma-3-12B exceeds 100% mCE in 9/10 categories (up to 102.3% on Blur), indicating worse robustness than the Molmo2-8B reference. The tight clustering suggests that on challenging reasoning tasks, relative robustness differences between models diminish.

*Table 28.* Mean Corruption Error (mCE, %) by model and corruption category. Lower is better; 100% matches the reference model. Reference models: Gemma-3-12B for MMBench, Molmo2-8B for MMMU-Pro. Categories match Table 1.

| MMBench (Reference: Gemma-3-12B, Baseline 85.3%) | | | | | | | | | | | |
|---|---|---|---|---|---|---|---|---|---|---|---|
| **Model** | **Base** | **Blur** | **Noise** | **Weath** | **Digi** | **Geom** | **Occl** | **Color** | **Resol** | **VLM** | **Bin** | **All** |
| Qwen3-VL-4B | 88.4 | 76.3 | 69.1 | 79.1 | 73.4 | 78.8 | 80.5 | 77.7 | 76.8 | 76.6 | 81.9 | 77.5 |
| Qwen3-VL-8B | 90.2 | 70.7 | 64.5 | 69.6 | 65.6 | 73.2 | 74.6 | 69.4 | 71.6 | 69.4 | 76.9 | 71.0 |
| Qwen3-VL-30B | 90.7 | 58.7 | 56.0 | 64.0 | 56.8 | 63.1 | 64.5 | 62.0 | 64.8 | 61.1 | 70.6 | 62.9 |
| InternVL3.5-4B | 86.3 | 101.1 | 93.6 | 99.0 | 96.0 | 98.8 | 99.1 | 96.7 | 99.7 | 94.9 | 100.8 | 98.3 |
| InternVL3.5-8B | 89.1 | 81.1 | 79.7 | 77.9 | 77.8 | 85.0 | 82.6 | 77.7 | 82.7 | 75.8 | 88.4 | 81.2 |
| InternVL3.5-14B | 86.6 | 93.2 | 86.0 | 91.1 | 89.3 | 91.1 | 94.2 | 91.7 | 92.2 | 86.4 | 98.1 | 92.0 |
| Molmo2-4B | 88.5 | 80.0 | 72.9 | 80.3 | 75.0 | 79.9 | 85.6 | 78.5 | 79.8 | 78.8 | 80.2 | 79.2 |
| Molmo2-8B | 88.4 | 79.5 | 70.1 | 76.9 | 77.5 | 80.5 | 81.0 | 77.8 | 81.9 | 75.1 | 79.6 | 78.2 |
| Qwen3-VL-32B | 92.4 | 58.9 | 53.1 | 56.5 | 55.5 | 59.0 | 61.8 | 55.7 | 61.3 | 55.0 | 66.1 | 58.6 |
| Qwen3-VL-235B | 93.4 | **52.0** | **44.9** | **45.4** | **45.1** | **51.9** | **53.5** | **46.8** | **54.4** | **43.5** | **57.9** | **50.0** |
| InternVL3.5-38B | 90.5 | 64.9 | 63.2 | 65.0 | 63.3 | 67.7 | 68.8 | 65.7 | 67.4 | 64.1 | 70.7 | 66.4 |
| Gemma-3-12B (ref) | 85.3 | 100.0 | 100.0 | 100.0 | 100.0 | 100.0 | 100.0 | 100.0 | 100.0 | 100.0 | 100.0 | 100.0 |
| GPT-5.4-mini | 87.7 | 85.0 | 88.5 | 89.1 | 79.7 | 84.3 | 85.1 | 84.5 | 86.5 | 80.0 | 90.0 | 85.9 |

| MMMU-Pro (Reference: Molmo2-8B, Baseline 31.2%) | | | | | | | | | | | |
|---|---|---|---|---|---|---|---|---|---|---|---|---|
| **Model** | **Base** | **Blur** | **Noise** | **Weath** | **Digi** | **Geom** | **Occl** | **Color** | **Resol** | **VLM** | **Bin** | **All** |
| Qwen3-VL-4B | 31.5 | 99.2 | 99.8 | 98.7 | 99.0 | 96.4 | 97.6 | 98.5 | 100.5 | 98.4 | 100.3 | 98.9 |
| Qwen3-VL-8B | 35.2 | 95.2 | 95.2 | 94.5 | 95.6 | 94.4 | 93.8 | 94.3 | 95.1 | 96.0 | 96.3 | 95.0 |
| Qwen3-VL-30B | 40.7 | 89.9 | 88.1 | 88.8 | 89.7 | 89.1 | 91.2 | 87.4 | 89.9 | 88.7 | 89.7 | 89.0 |
| InternVL3.5-4B | 37.3 | 94.8 | 92.9 | 91.7 | 92.5 | 92.4 | 94.7 | 91.6 | 93.9 | 93.7 | 94.6 | 93.1 |
| InternVL3.5-8B | 41.0 | 91.6 | 88.8 | 88.0 | 90.0 | 88.7 | 91.2 | 87.6 | 89.0 | 87.0 | 90.6 | 89.1 |
| InternVL3.5-14B | 42.0 | 86.6 | 84.8 | 83.9 | 85.5 | 84.2 | 85.8 | 83.8 | 86.0 | 84.4 | 86.4 | 85.0 |
| Molmo2-4B | 31.8 | 99.2 | 100.7 | 100.4 | 99.6 | 97.6 | 99.1 | 100.4 | 99.7 | 100.7 | 101.5 | 100.0 |
| Molmo2-8B (ref) | 31.2 | 100.0 | 100.0 | 100.0 | 100.0 | 100.0 | 100.0 | 100.0 | 100.0 | 100.0 | 100.0 | 100.0 |
| Gemma-3-12B | 33.0 | 102.3 | 101.8 | 101.1 | 100.2 | 99.5 | 101.9 | 100.1 | 101.6 | 101.0 | 101.9 | 101.1 |
| Qwen3-VL-32B | 43.2 | 84.9 | 84.4 | 84.2 | **84.7** | 85.4 | 85.0 | 83.1 | 85.9 | 84.0 | 86.3 | 84.7 |
| Qwen3-VL-235B | 46.0 | **80.2** | **79.2** | **81.8** | 89.6 | **81.8** | **79.7** | **77.8** | **80.7** | **78.3** | **80.2** | **80.2** |
| InternVL3.5-38B | 41.4 | 87.2 | 86.1 | 86.1 | 87.9 | 85.2 | 86.9 | 85.4 | 87.2 | 86.7 | 88.0 | 86.5 |
| GPT-5.4-mini | 39.2 | 93.1 | 91.4 | 89.9 | 90.4 | 90.3 | 91.5 | 88.6 | 91.7 | 89.8 | 91.2 | 90.6 |

# C. Experiment Details

This section provides implementation details for reproducibility.

## C.1. Random Seeds

We use fixed random seeds throughout all experiments to ensure reproducibility:

- **Sampling seed**: 42 — used for stratified dataset sampling to select the 20% evaluation subset.

- **Augmentation seed**: 1234 — base seed for deterministic per-sample augmentation. Each sample $i$ receives seed $(1234 \times 1000003 + i) \mod 2^{32}$ to ensure reproducible yet varied augmentations.

- **Generation seed**: 42 — used for thinking-mode models that require sampling-based decoding.

## C.2. Dataset Sampling

To reduce computational costs while maintaining statistical validity, we evaluate on a 20% stratified subset of each benchmark:

- **MMBench**: 853 samples from 4,329 total (stratified by `category` field)

- **MMMU-Pro**: 324 samples from 1,730 total (stratified by `subject` field)

Stratified sampling ensures proportional representation of all question categories/subjects in the evaluation subset.

## C.3. Prompting Templates

We use two prompting modes with standardized templates:

**Direct Mode.**  Designed for short, single-letter responses:

```
Please select the correct answer from the options above.  Respond with only the
letter of the correct option.  Do not explain.  Answer:
```

**Chain-of-Thought (CoT) Mode.**  Designed for reasoning-based responses:

```
Answer the preceding multiple choice question.  The last line of your response
should be of the following format:  'Answer: $LETTER' (without quotes) where
LETTER is one of options.  Think step by step before answering.
```

## C.4. Generation Parameters

All models use deterministic decoding with `max_new_tokens=2048`. Thinking models (Qwen3-VL-Thinking) require sampling-based decoding and use: `max_new_tokens=8192`, `temperature=0.6`, `top_p=0.95`, `top_k=20`.

## C.5. Augmentation Parameters

Table 29 lists the parameter values for each severity level across all severity-based augmentations. We evaluate at low, mid, and high severities. Binary augmentations have no severity variation.

**Binary Augmentations.**  The following 7 augmentations have no severity levels: `flip_h`, `flip_v`, `grayscale`, `invert`, `channel_swap`, `equalize`, `autocontrast`.

## C.6. Augmentation Application

Augmentations are applied deterministically based on sample index:

1. For each sample index $i$, compute per-sample seed: $s_i = (1234 \times 1000003 + i) \mod 2^{32}$

2. Initialize augmentation RNG with $s_i$

3. Apply augmentation to all images in the sample

This ensures: (1) identical augmentation across model runs for fair comparison, (2) different random variations per sample for stochastic augmentations (noise, blur, occlusion positions).

## C.7. Evaluation Protocol

**Correctness.**  A response is correct if the extracted letter matches the ground truth answer field.

**Metrics.**  All metrics (accuracy, flip rates, RCE, mCE) are computed on the same 20% stratified subset across all models and augmentations, enabling direct comparison.

## D. Per-Category Robustness Analysis

Table 30 reports per-category robustness on MMBench averaged across the 12 open-weights models. Worst-case drops range from 24 pp to 49 pp across all categories, confirming that spatial fragility is not confined to a single task type. OCR is the most vulnerable category, with a distinct vulnerability pattern: `elastic_transform` dominates (not `upsample`), as geometric distortion destroys text readability.

Table 31 reports per-domain robustness on MMMU-Pro.

## E. Flip Ground-Truth Validity Details

We used GPT-5.4 as a judge to classify each question as spatially sensitive (i.e., requiring knowledge of left-right or top-bottom orientation to answer correctly). Only 3–6% of questions are inherently spatially sensitive (47/853 MMBench, 10/324 MMMU-Pro). Table 32 re-computes flip drops after excluding all sensitive questions.

*Table 29.* Augmentation parameters by severity level. Values at low, mid, and high severity are used in experiments.

| Category | Augmentation | Param | Low | Mid | High | Note |
|---|---|---|---|---|---|---|
| Blur | gaussian_blur | radius | 0.5 | 1.5 | 2.5 | pixels |
| | motion_blur | ksize | 5 | 9 | 15 | kernel size |
| | defocus_blur | radius | 1.0 | 3.0 | 5.0 | pixels |
| | zoom_blur | factor | 0.02 | 0.06 | 0.10 | zoom amount |
| | glass_blur | sigma | 0.5 | 0.9 | 1.3 | blur sigma |
| Noise | gaussian_noise | std | 0.02 | 0.06 | 0.10 | normalized |
| | shot_noise | scale | 25 | 10 | 5 | lower=more |
| | speckle_noise | std | 0.05 | 0.15 | 0.25 | normalized |
| | salt_pepper | amount | 0.01 | 0.04 | 0.08 | pixel fraction |
| Weather | fog | intensity | 0.2 | 0.6 | 1.0 | opacity |
| | frost | intensity | 0.2 | 0.6 | 1.0 | opacity |
| | snow | intensity | 0.1 | 0.3 | 0.5 | density |
| | rain | intensity | 0.1 | 0.3 | 0.5 | density |
| | spatter | intensity | 0.1 | 0.3 | 0.5 | coverage |
| Digital | jpeg_compression | quality | 80 | 50 | 20 | lower=worse |
| | pixelate | scale | 0.9 | 0.5 | 0.2 | lower=coarser |
| Geometric | rotate | degrees | 5 | 15 | 30 | rotation |
| | shear | degrees | 5 | 15 | 25 | shear angle |
| | affine | degrees | 5 | 15 | 30 | rotation+scale |
| | perspective_transform | magnitude | 0.05 | 0.15 | 0.25 | distortion |
| | elastic_transform | alpha | 30 | 80 | 180 | deformation |
| Color/Tone | brightness | factor | 0.7 | 0.3 | 0.1 | lower=darker |
| | brightness_up | factor | 1.3 | 1.7 | 2.5 | higher=brighter |
| | contrast | factor | 0.7 | 0.3 | 0.1 | lower=flatter |
| | contrast_up | factor | 1.3 | 1.8 | 3.0 | higher=sharper |
| | saturation | factor | 0.5 | 0.1 | 0.0 | lower=grayer |
| | saturation_up | factor | 1.5 | 2.5 | 4.0 | higher=vivid |
| | gamma | factor | 0.7 | 0.4 | 0.2 | lower=brighter |
| | gamma_up | factor | 1.3 | 2.0 | 3.0 | higher=darker |
| | hue_shift | degrees | 10 | 40 | 90 | color rotation |
| | color_jitter | range | 0.1 | 0.3 | 0.5 | random B/C/S |
| Occlusion | random_occlusion | ratio | 0.05 | 0.15 | 0.25 | area blocked |
| | grid_mask | ratio | 0.1 | 0.2 | 0.3 | grid density |
| | center_occlusion | ratio | 0.1 | 0.3 | 0.5 | center blocked |
| Resolution | downsample | scale | 0.75 | 0.35 | 0.15 | lower=smaller |
| | upsample | scale | 1.5 | 3.0 | 6.0 | interpolation |
| | sharpen | factor | 1.5 | 3.0 | 6.0 | edge enhance |
| | posterize | bits | 6 | 4 | 2 | lower=fewer |
| | solarize | threshold | 200 | 128 | 64 | lower=more |
| VLM-specific | text_overlay | fontsize | 24 | 48 | 72 | pixels |
| | watermark | fontsize | 24 | 48 | 72 | pixels |
| | add_border | width | 10 | 30 | 60 | pixels |

| MMBench Category | Clean Acc | Worst Drop | Worst Aug |
|---|---|---|---|
| OCR | 98.7% | 49.2pp | elastic_transform (high) |
| relation_reasoning | 91.2% | 39.9pp | upsample (high) |
| finegrained_perception (inst.) | 89.9% | 36.3pp | upsample (high) |
| logic_reasoning | 85.5% | 28.4pp | upsample (high) |
| attribute_reasoning | 90.5% | 31.6pp | upsample (high) |
| coarse_perception | 90.4% | 24.2pp | upsample (high) |

*Table 30.* Per-category robustness on MMBench (12 open-weights model average). Worst Drop = maximum, across the 133 corruption configurations, of the cross-model mean drop. OCR is the most fragile category and the only one whose worst-case is driven by `elastic_transform`.

| MMMU-Pro Domain | Clean Acc | Worst Drop | Worst Aug |
|---|---|---|---|
| Art & Design | 56.0% | 29.7pp | zoom_blur (high) |
| Humanities & Social Science | 46.6% | 10.9pp | elastic_transform (high) |
| Health & Medicine | 28.2% | 8.5pp | upsample (high) |
| Business | 32.3% | 8.3pp | elastic_transform (high) |
| Science | 40.1% | 7.7pp | elastic_transform (high) |
| Tech & Engineering | 32.1% | 5.4pp | zoom_blur (mid) |

*Table 31.* Per-domain robustness on MMMU-Pro (12 open-weights model average). All domains show non-trivial worst-case drops, with spatial/resampling corruptions as the universal worst case.

| Dataset | Flip | All Qs | Non-Sensitive Only | Drop Retained |
|---|---|---|---|---|
| MMBench | H-flip | 7.0pp | 6.3pp | 90% |
| MMBench | V-flip | 10.2pp | 9.9pp | 97% |
| MMMU-Pro | H-flip | 3.6pp | 3.6pp | 100% |
| MMMU-Pro | V-flip | 4.8pp | 4.8pp | 100% |

*Table 32.* Flip-induced drops before and after excluding spatially sensitive questions. 90–100% of drops are retained, confirming they reflect model fragility rather than ground-truth invalidation.

# F. Augmentation Visualization

Figures 3–14 visualize all 49 augmentations applied to a representative MMBench image at low, mid, and high severity levels. Binary augmentations have no severity variation.

# Blur

| Low | Mid | High |
|-----|-----|------|

Gaussian Blur

Motion Blur

Defocus Blur

Zoom Blur

Glass Blur

*Figure 3.* Augmentation Visualization: Blur augmentations at low, mid, and high severity.

# Noise

*Figure 4.* Augmentation Visualization: Noise augmentations at low, mid, and high severity.

# Weather

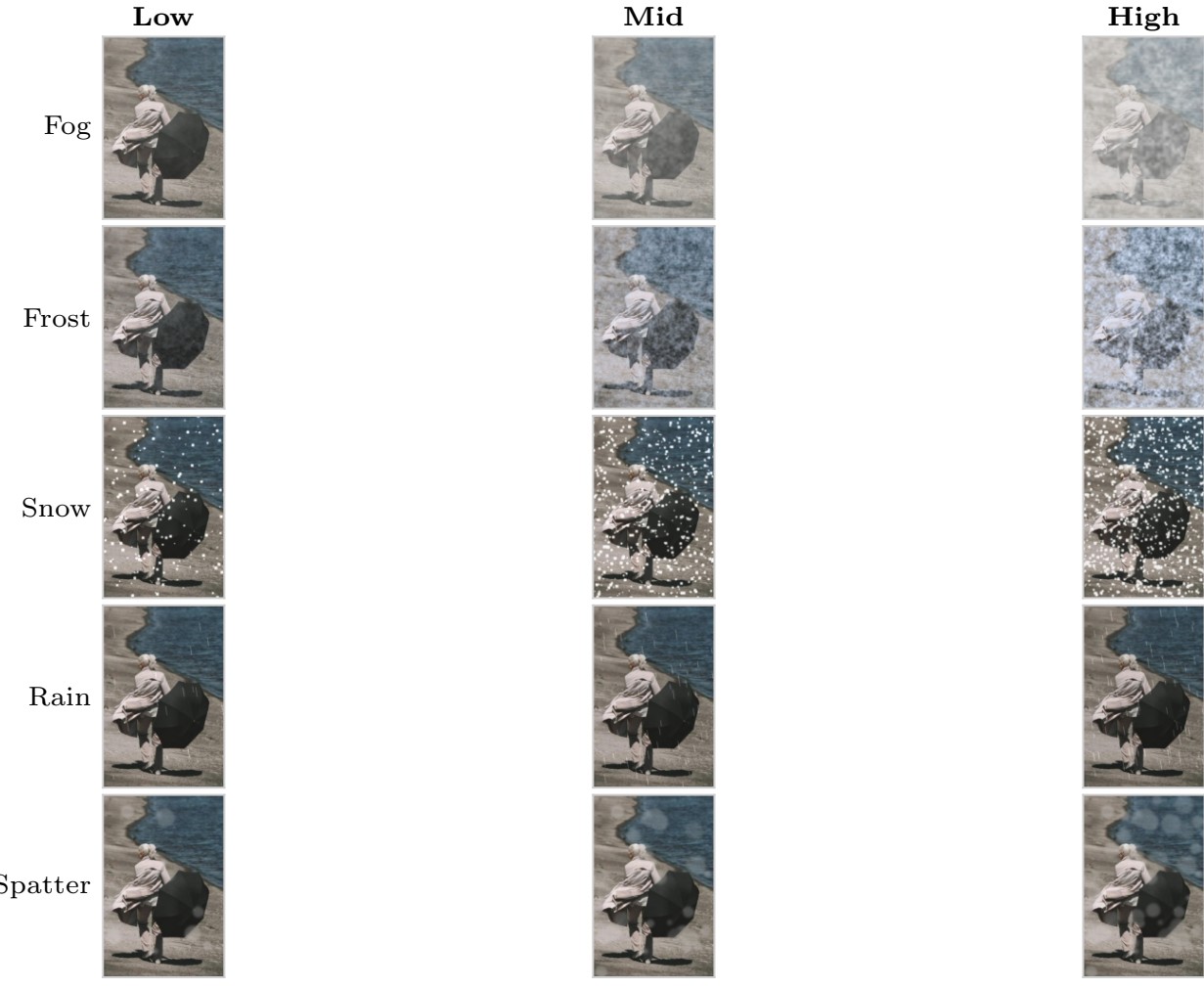

*Figure 5.* Augmentation Visualization: Weather augmentations at low, mid, and high severity.

## Digital

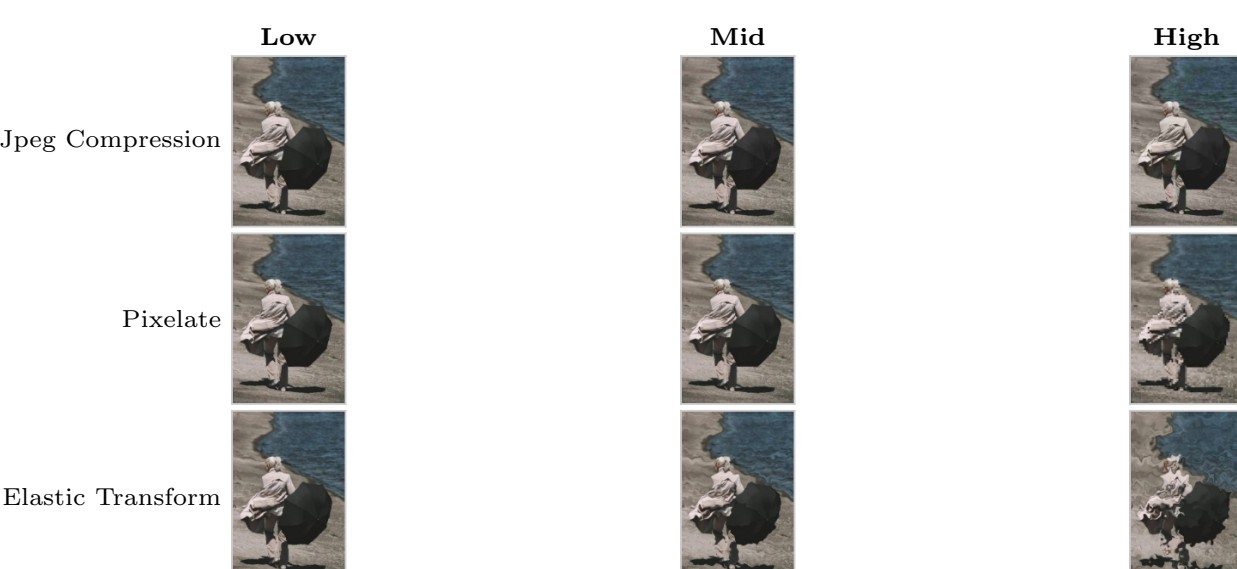

*Figure 6.* Augmentation Visualization: Digital augmentations at low, mid, and high severity.

## Geometric

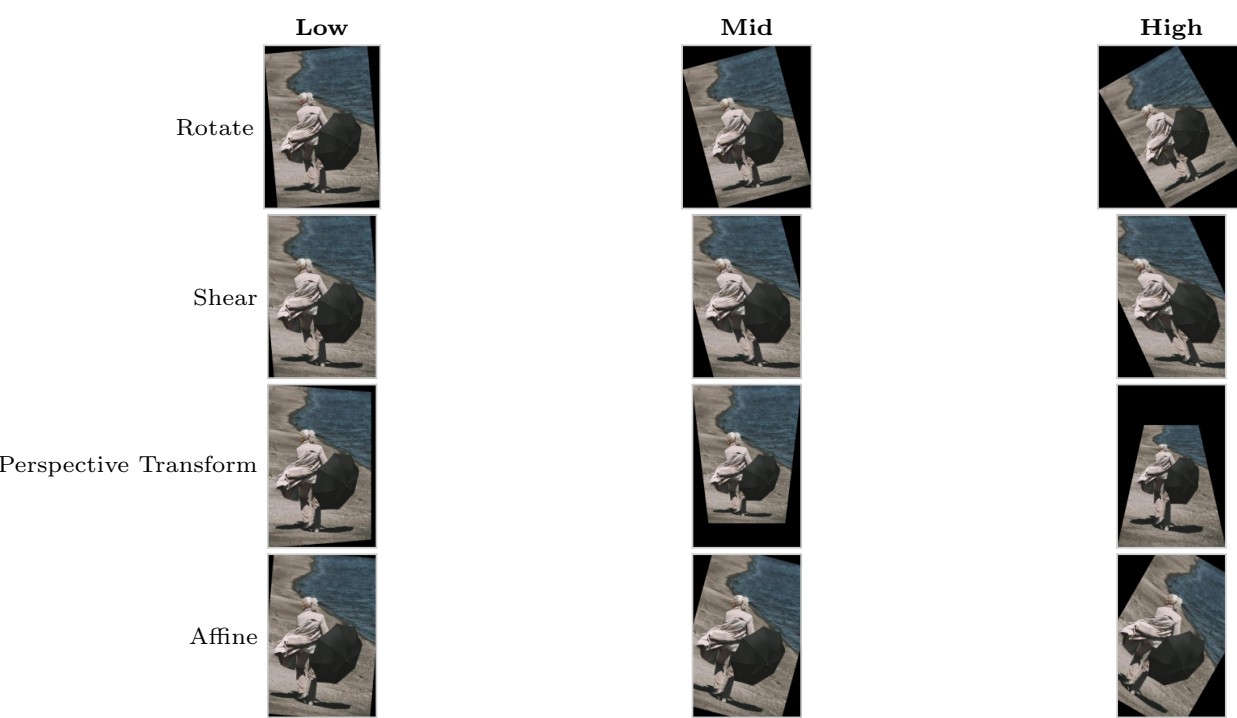

*Figure 7.* Augmentation Visualization: Geometric augmentations at low, mid, and high severity.

# Color (Decrease)

*Figure 8.* Augmentation Visualization: Color/Tone decrease augmentations at low, mid, and high severity.

# Color (Increase)

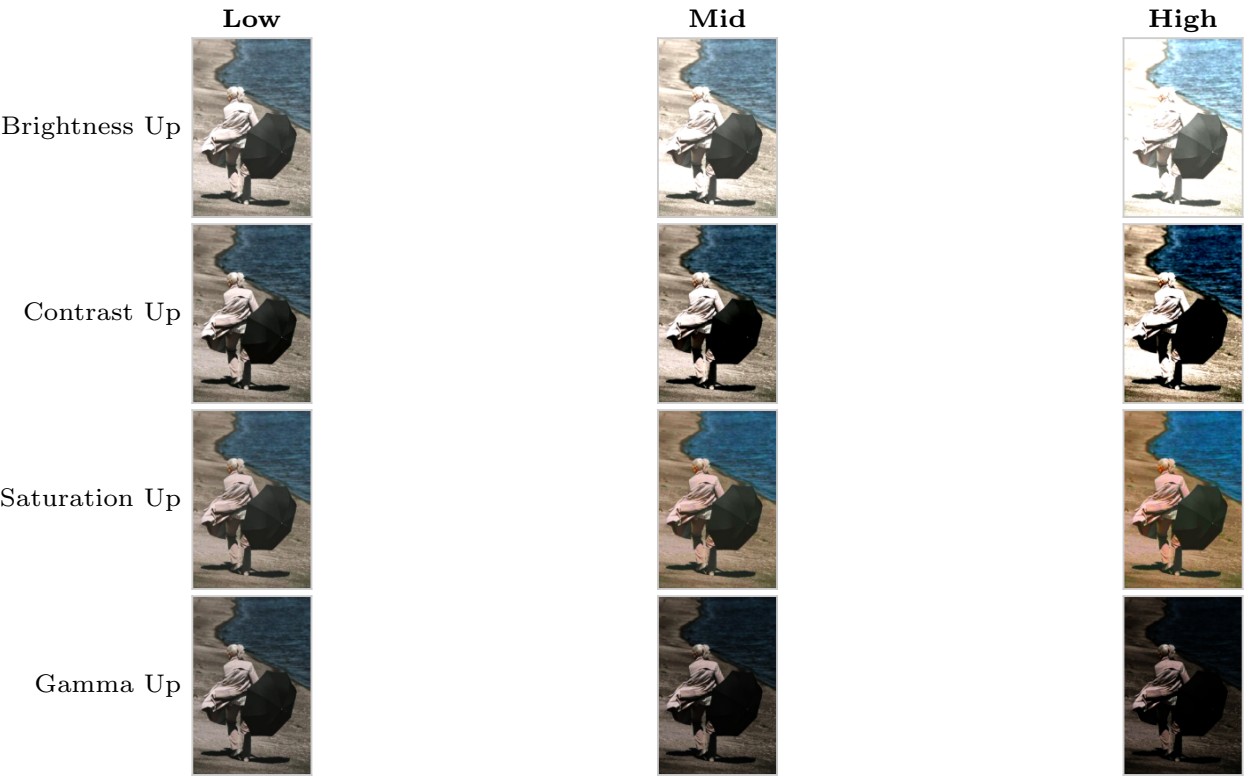

*Figure 9.* Augmentation Visualization: Color/Tone increase augmentations at low, mid, and high severity.

# Color (Other)

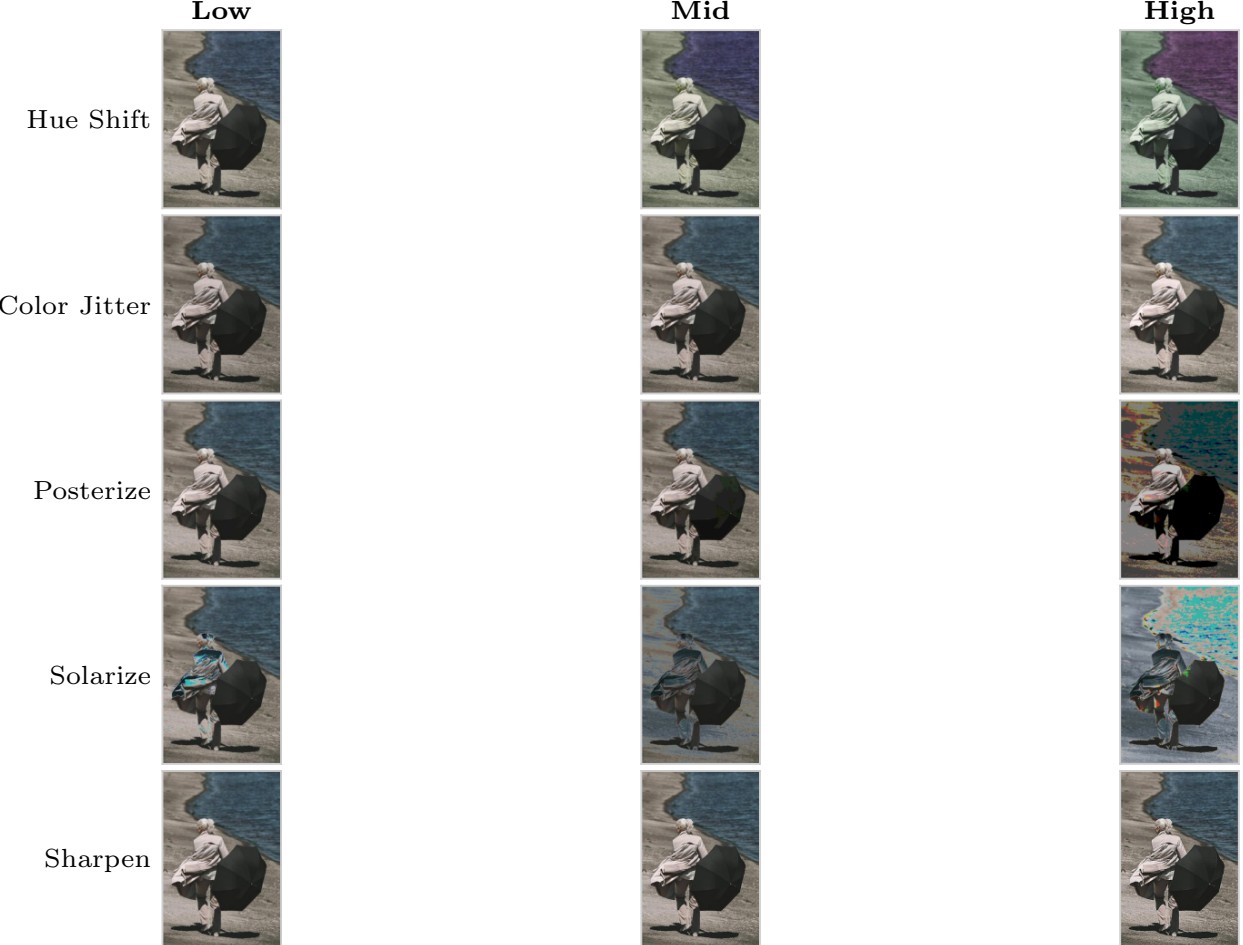

*Figure 10.* Augmentation Visualization: Other Color/Tone augmentations at low, mid, and high severity.

# Occlusion

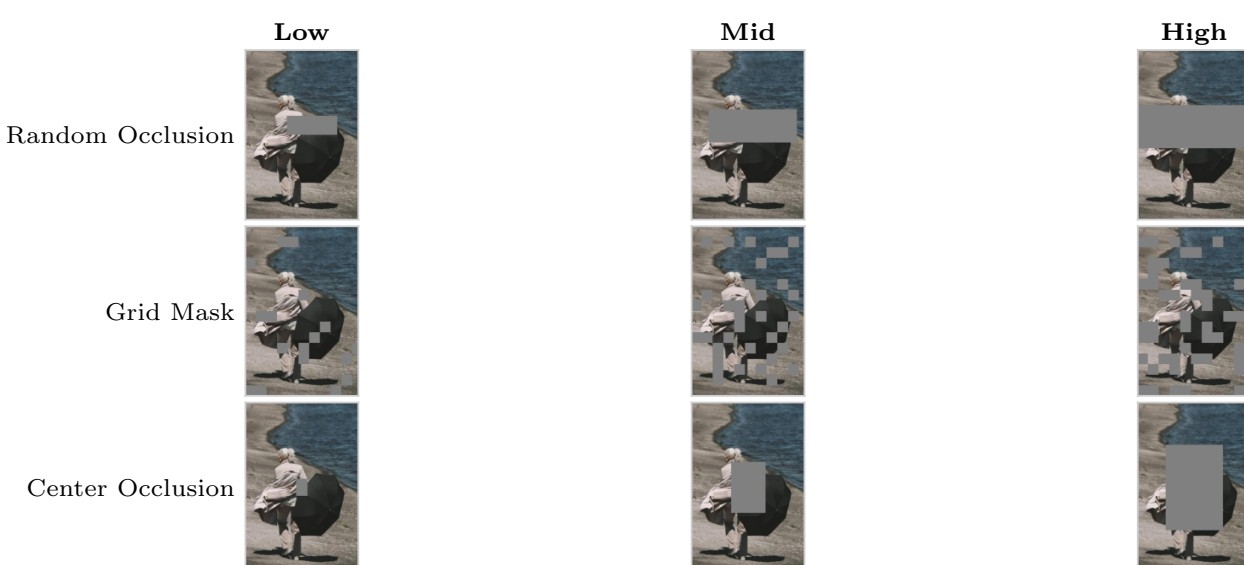

*Figure 11.* Augmentation Visualization: Occlusion augmentations at low, mid, and high severity.

# Resolution

*Figure 12.* Augmentation Visualization: Resolution augmentations at low, mid, and high severity.

# VLM-Specific

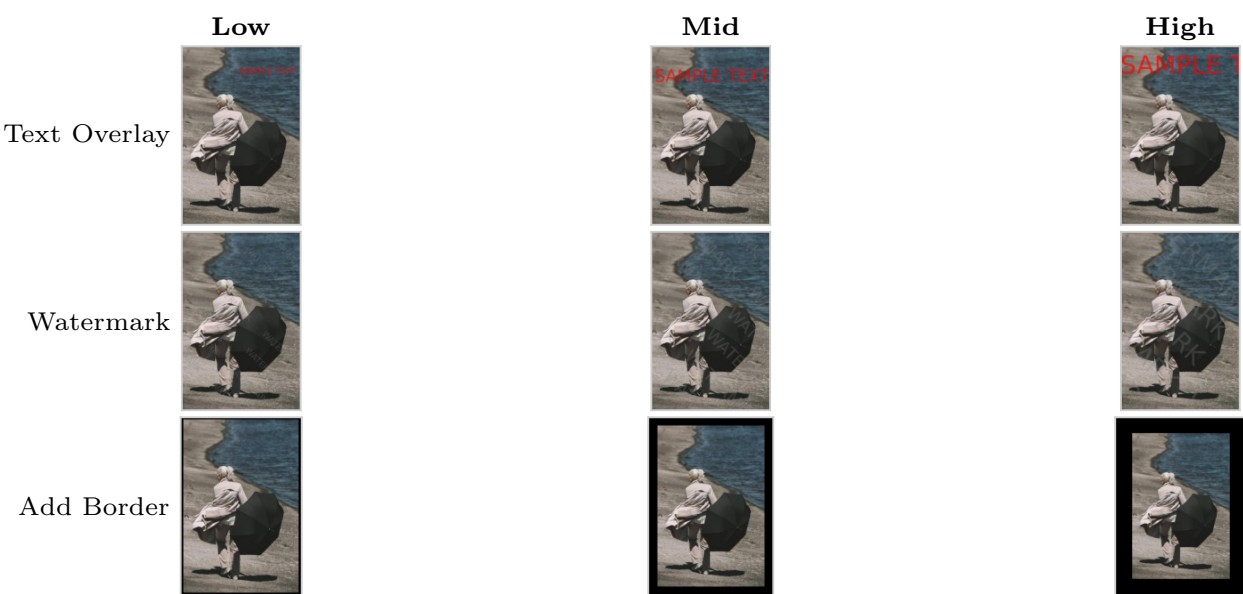

*Figure 13.* Augmentation Visualization: VLM-specific augmentations at low, mid, and high severity.

# Binary Transforms

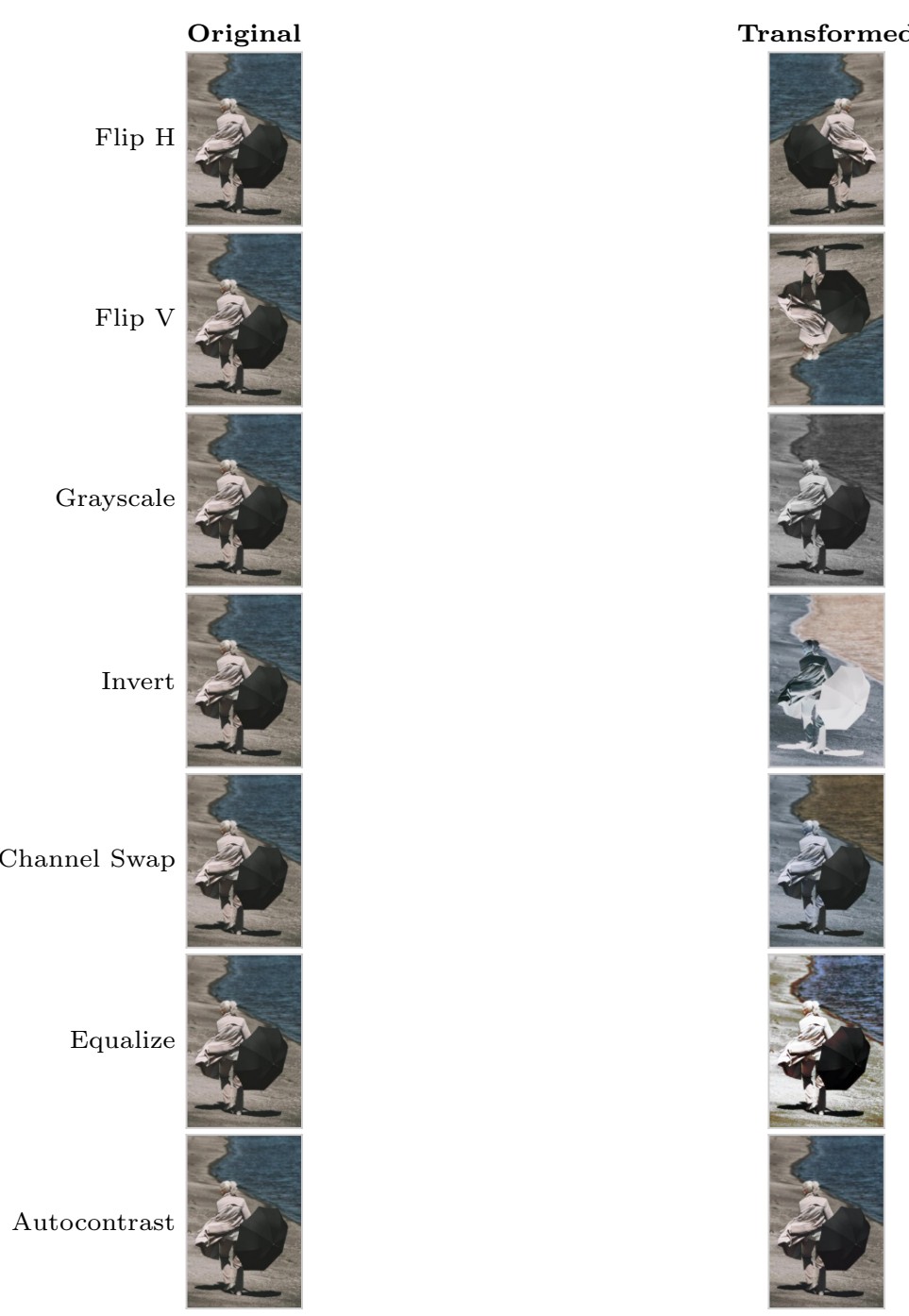

*Figure 14.* Augmentation Visualization: Binary transforms (no severity variation).

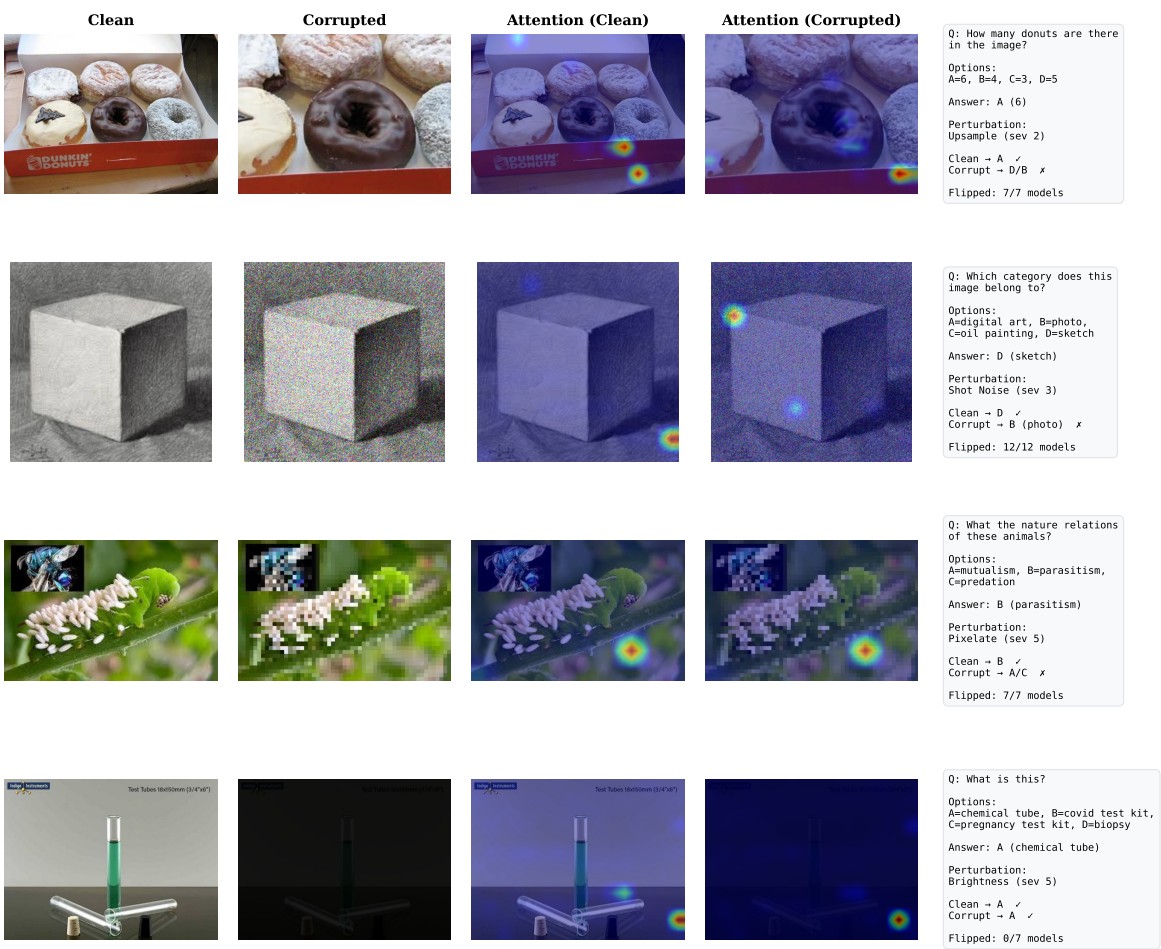

*Figure 15.* Qualitative examples of augmentation effects on VLM predictions. Rows 1–3 show failure cases where all tested models flip from correct to incorrect after corruption, despite the corrupted image remaining answerable by humans. Row 4 shows a robustness case: extreme brightness reduction (factor 0.1) produces a nearly black image, yet all models maintain the correct answer.

