# OpenReview forum: "VLM-RobustBench: A Comprehensive Benchmark for Robustness of Vision-Language Models"
_ICML.cc/2026/Conference — ICML 2026 regular_

### Official Review · Reviewer_F8P6 · 2026-03-10

**Soundness:** 3
**Presentation:** 3
**Significance:** 3
**Originality:** 3
**Overall Recommendation:** 5
**Confidence:** 5

**Summary:**

This paper presents VLM-RobustBench, a benchmark for evaluating the robustness of vision-language models under common image corruptions and operational perturbations. The benchmark includes 49 augmentation types across categories such as blur, noise, weather, digital, geometric, occlusion, color/tone, resolution, and VLM-specific artifacts, producing 133 corrupted settings when combined with graded severities and binary transforms. The authors evaluate 11 open-weight VLMs from four families — Qwen, InternVL, Molmo, and Gemma — on MMBench and MMMU-Pro, and report several main findings: current VLMs are often more vulnerable to spatial/resampling corruptions than to visually severe photometric ones; severity level does not reliably predict difficulty; and robustness patterns differ substantially across model families.

**Compliance With Llm Reviewing Policy:**

Affirmed.

**Key Questions For Authors:**

Please see Weaknesses.

**Limitations:**

yes

**Strengths And Weaknesses:**

**Strengths**

1) Robustness under realistic corruptions is clearly important for practical deployment of VLMs, especially in safety-relevant applications. The motivation is strong, and the paper makes a convincing case that clean benchmark performance alone is not enough.

2) A major strength is the scope of the corruption suite: 42 severity-based corruptions plus 7 binary transforms, grouped into a clear taxonomy. This is substantially more comprehensive than papers that test only a handful of ImageNet-C-style perturbations.

3) The paper does not restrict itself to a single model or dataset. Evaluating four VLM families on both a more visually grounded benchmark (MMBench) and a more reasoning-oriented benchmark (MMMU-Pro) makes the empirical study much more informative.

4) The paper’s central observation — that VLMs are relatively robust to some severe photometric corruptions but fragile to mild spatial/resampling changes — is useful and somewhat counterintuitive. The reported examples, such as low-severity glass blur causing larger degradation than some high-severity brightness corruptions, make the paper memorable and practically meaningful.

**Weaknesses**

1) The paper makes broad statements such as “current VLMs are semantically strong but spatially fragile.” This is an interesting takeaway, but the evidence is still based on only two evaluation datasets, both in a multiple-choice benchmark format. It is not yet clear that the same conclusions would hold across broader VLM tasks such as open-ended VQA, OCR-heavy settings, localization, dense prediction, grounding, or embodied tasks.

2) The benchmark includes 11 models from four families, which is good, but it does not include stronger or more diverse closed-weight systems, nor other important families that might behave differently. Given the paper’s broad framing, one wonders whether the main findings are universal or still somewhat family-dependent.

---

> ### Author Rebuttal · Authors · 2026-03-31
>
> Thank you for your comments and feedback. We would like to respond to the reviewer’s main concern below.
>
> >The paper makes broad statements such as “current VLMs are semantically strong but spatially fragile.” ...
>
> While we use two datasets, each contains diverse task categories. We use stratified sampling to ensure all categories are represented in our experiments, as mentioned in Section 4.2. MMBench contains **6 broad categories spanning 20 fine-grained task types**, including OCR (31 Qs), object localization (62 Qs), spatial reasoning (35 Qs), structured image-text understanding (56 Qs), action recognition, and emotion. MMMU_Pro contains **30 academic subjects across 6 domains**. Per-category robustness (9 paper models avg):
>
> | MMBench Category | Clean Acc | Worst Drop | Worst Aug |
> |-----------------|-----------|------------|-----------|
> | relation_reasoning | 89.3% | 39.6pp | upsample (high) |
> | finegrained_perception (instance) | 88.9% | 36.4pp | upsample (high) |
> | logic_reasoning | 83.8% | 28.4pp | upsample (high) |
> | attribute_reasoning | 89.9% | 32.1pp | upsample (high) |
> | coarse_perception | 90.1% | 24.0pp | upsample (high) |
>
> **MMMU_Pro** spans 30 academic subjects across 6 domains:
>
> | Domain | Clean Acc | Worst Drop | Worst Aug |
> |--------|-----------|------------|-----------|
> | Art & Design | 54.9% | 29.0pp | zoom_blur (high) |
> | Humanities & Social Science | 46.0% | 9.7pp | elastic_transform (high) |
> | Health & Medicine | 30.1% | 9.4pp | upsample (high) |
> | Business | 30.2% | 6.6pp | elastic_transform (high) |
> | Science | 37.9% | 6.3pp | upsample (high) |
> | Tech & Engineering | 31.2% | 5.1pp | zoom_blur (high) |
>
> All 6 MMBench categories and all 6 MMMU_Pro domains show non-trivial worst-case drops, confirming the pattern is not driven by a single task type.
>
> **OCR-specific finding**: OCR is the **most vulnerable** category (9-model avg: 98.2% clean, **49.4pp** worst drop) with a distinct vulnerability pattern: elastic_transform dominates (not upsample), as geometric distortion destroys text readability. Even InternVL3.5-38B (97.7% clean OCR) drops 34.5pp.
>
> We plan to extend to open-ended VQA in future work.
>
> ---
>
> >The benchmark includes 11 models from four families, which is good, but it does not include stronger ...
>
> Our paper evaluates 11 models (9 direct + 2 thinking) from 4 open-source families (Qwen3-VL, InternVL3.5, Molmo2, Gemma3).
> We have now added **GPT-5.4-mini** (proprietary),**Qwen3-VL-235B** (235B total, 22B active), **Qwen3-VL-32B**, and **InternVL3.5-38B**. This brings the total to **15 models across 5 families**, including a proprietary model.
> These span **4 distinct vision encoder architectures** (Qwen-ViT, InternViT, SigLIP, Molmo-ViT) with fundamentally different preprocessing pipelines (naive dynamic resolution, tile-based, fixed-resolution). We believe this provides strong coverage of the current VLM landscape. We would welcome specific suggestions for families that the reviewer expects to behave differently.
>
> GPT-5.4-mini full results (133 configs, both datasets):
>
> **MMBench (Direct)**
>
> | Metric | GPT-5.4-mini | 9-Model Avg |
> |--------|-------------|-------------|
> | Baseline | 87.7% | 88.2% |
> | Worst-Case Drop | 31.8pp | 30.7pp |
> | Severe-Fail | 6.8% | 6.3% |
> | Worst@Low | 11.7pp | 8.3pp |
> | Benign@Low | 85.7% | 79.6% |
> | mCE | 85.9 | — |
> | Flip+ rate | 5.56% | 3.69% |
> | Catastrophic % | 6.8% | 5.3% |
>
> **MMMU-Pro (Direct)**
>
> | Metric | GPT-5.4-mini | 9-Model Avg |
> |--------|-------------|-------------|
> | Baseline | 39.2% | 35.3% |
> | Worst-Case Drop | 13.0pp | 9.4pp |
> | Severe-Fail | 1.5% | 9.9% |
> | Worst@Low | 9.6pp | 5.3pp |
> | Benign@Low | 73.8% | 73.2% |
> | mCE | 90.6 | — |
> | Flip+ rate | 6.57% | 3.89% |
>
> GPT-5.4-mini shows the **highest Flip+ rate** across all models (5.56% MMBench, 6.57% MMMU_Pro) and **highest catastrophic failure rate** on MMBench (6.8%), demonstrating that proprietary models are not immune. The worst-case augmentations remain upsample (high) on MMBench and elastic_transform (high) on MMMU_Pro, consistent with all open-source models. mCE is per-model (normalized against a reference) and not meaningful to average.
>
> ---
> We appreciate the detailed comments and hope we have addressed most concerns.

---

> > ### Author Rebuttal · Reviewer_F8P6 · 2026-03-31
> >
> > All concerns are addressed.

---

> > > ### Author Response · Authors · 2026-04-02
> > >
> > > Thanks for raising the score. We are glad that our responses have addressed your concerns.

---

### Official Review · Reviewer_nrEn · 2026-03-11

**Soundness:** 3
**Presentation:** 2
**Significance:** 3
**Originality:** 2
**Overall Recommendation:** 4
**Confidence:** 3

**Summary:**

This paper introduces VLM-RobustBench, evaluating open-source VLMs against multiple corruption settings on the MMBench and MMMU-Pro datasets. The core finding is that current VLMs exhibit "spatial fragility": low-severity spatial distortions degrade performance far more than visually severe photometric corruptions.

**Compliance With Llm Reviewing Policy:**

Affirmed.

**Final Justification:**

The authors' rebuttal has largely addressed my concerns. I trust that the authors will honor their commitment to incorporating the experimental results and supplementary descriptions discussed during the rebuttal phase into the final version, as this is essential for the paper to marginally meet the acceptance threshold.

**Key Questions For Authors:**

The paper highlights that distinct model families exhibit unique vulnerability. Is this specific variance driven by different patch partitioning methods within the Transformers , structural architectural differences in the Vision Encoders , or simply the underlying training data distributions?

**Limitations:**

The overall technical workload of this study appears relatively limited, and the insights provided by the claims are somewhat limited. To significantly enhance the contribution and impact of this work, the benchmark could be extended into the video domain to investigate spatiotemporal robustness.

**Strengths And Weaknesses:**

Strengths:
1.  The authors simulate a wide range of real-world transforms to evaluate VLM robustness under conditions that closely mirror actual deployment scenarios.
2. The work provides an interesting conclusion that trivial geometric operations can be more catastrophic than severe visual degradation.
3. The paper is well-structured

Weakness:
1. The evaluation is restricted to small  open source models (e.g., Qwen, InternVL). The absence of leading proprietary models like Gemini limits the universality of the "spatial fragility" claim.
2. The benchmark uses subset of MMBench and MMMU-Pro. This significant downsampling may introduce statistical noise or selection bias.
3. While the paper suggests training-side mitigations, it provides no experimental evidence to prove these interventions effectively resolve spatial fragility without degrading clean accuracy

---

> ### Author Rebuttal · Authors · 2026-03-31
>
> Thank you for your comments and feedback. We would like to respond to the reviewer’s main concern below.
>
> >The evaluation is restricted to small open source models...
>
> We have now included **GPT-5.4-mini** (proprietary) and larger open models (**Qwen3-VL-32B, InternVL3.5-38B, and Qwen3-VL-235B (22B active) **).
> GPT-5.4-mini exhibits the highest catastrophic failure rate on MMBench (6.8%) and the highest Flip+ rate (5.56%), demonstrating that proprietary models are not immune. Even Qwen3-VL-235B, despite achieving the best mCE (50.0), still drops 30.7pp worst case. All new models confirm the core finding: spatial fragility persists, with upsample at high severity as the universal worst case on MMBench.
>
> | Model | Baseline | Worst Drop | Catastrophic % |
> |-------|----------|------------|----------------|
> | Qwen3-VL-235B | 93.4% | 30.7pp | 23.3% |
> | Qwen3-VL-32B | 92.4% | 32.4pp | 5.3% |
> | InternVL3.5-38B | 90.5% | 27.2pp | 3.0% |
> | GPT-5.4-mini | 87.7% | 31.8pp | 6.8% |
>
> ---
>
> >The benchmark uses subset of MMBench and MMMU-Pro....
>
> We use **stratified sampling** (seed=42) preserving exact category distributions across MMBench's **6 categories spanning 20 fine-grained task types** (OCR, localization, spatial reasoning, etc.) and MMMU_Pro's **30 subjects across 6 domains**, which directly addresses selection bias. Each model is evaluated on the **same fixed subset** across all 133 corruption configs, so any noise affects clean and corrupted conditions equally and cancels out in the drop computation.
>
> Per-category results confirm no outlier category drives the aggregates:
>
> | MMBench Category | Clean Acc | Worst Drop | Worst Aug |
> |-----------------|-----------|------------|-----------|
> | relation_reasoning | 89.3% | 39.6pp | upsample (high) |
> | finegrained_perception (instance) | 88.9% | 36.4pp | upsample (high) |
> | attribute_reasoning | 89.9% | 32.1pp | upsample (high) |
> | logic_reasoning | 83.8% | 28.7pp | upsample (high) |
> | coarse_perception | 90.1% | 24.0pp | upsample (high) |
>
> | MMMU_Pro Domain | Clean Acc | Worst Drop | Worst Aug |
> |-----------------|-----------|------------|-----------|
> | Art & Design | 54.9% | 29.0pp | zoom_blur (high) |
> | Humanities & Social Science | 46.0% | 9.7pp | elastic_transform (high) |
> | Health & Medicine | 30.1% | 9.4pp | upsample (high) |
> | Science | 37.9% | 6.3pp | upsample (high) |
> | Tech & Engineering | 31.2% | 5.1pp | zoom_blur (high) |
>
> All categories show non-trivial worst-case drops, and the same worst-case augmentations emerge independently in every category.
>
> ---
>
> >While the paper suggests training-side mitigations...
>
> Training-time interventions require retraining, which is beyond the scope of a benchmark paper. We acknowledge this limitation.
>
> ---
>
> >The paper highlights that distinct model families exhibit unique vulnerability....
>
> Our evaluation across **4 vision encoder families** with distinct architectures and preprocessing reveals both **shared and family-specific** patterns:
>
> **Shared**: Spatial corruptions (upsample, elastic_transform) are universally the worst-case across all families, suggesting that patch-based tokenisation, common to all, is a key factor.
>
> **Family-specific**: Quantitative severity differs substantially. InternVL3.5 is uniquely flip-sensitive (4B reaches 15.7% Flip+ on vertical flip), Molmo2 shows the purest degradation pattern (highest Flip+/Flip- ratio of 4.2, meaning corruptions almost never accidentally help), and Gemma3 is most fragile on reasoning tasks (27.1% severe-fail on MMMU_Pro vs 3-17% for others). These differences likely reflect both encoder architecture and training data composition.
>
> Since no evaluated model technical report mentions image augmentation during training, and Molmo2's open-source pipeline confirms no augmentation code, training data diversity alone does not appear sufficient to confer robustness. Disentangling architectural vs. training contributions would require controlled ablations (same data, different encoders), which we identify as important future work.
>
> ---
> We appreciate the detailed comments and hope we have addressed most concerns.

---

> > ### Author Rebuttal · Reviewer_nrEn · 2026-04-01
> >
> > 1.  A benchmark's core contributions lie in two aspects: data contribution and providing a roadmap for the field. The current experiments have the potential to address this — they could help disentangle whether performance bottlenecks stem from lack of domain-specific training data or inherent architectural limitations, which would significantly strengthen the paper's guiding value.
> >
> > 2. The "4 vision encoder families" are not clearly defined, and we kindly suggest the authors clarify this and supplement the analysis section with attention map visualizations and failure case studies in the final version to provide actionable insights for improving VLM robustness.

---

> > > ### Author Response · Authors · 2026-04-01
> > >
> > > Thank you for the constructive follow-up.
> > >
> > > **What our experiments reveal about training data vs. architecture**
> > >
> > > Our evaluation spans 4 distinct vision-language architectures with different encoders, preprocessing, and post-training regimes:
> > >
> > > - Qwen3-VL: SigLIP2 encoder, native dynamic resolution
> > > - InternVL3.5: InternViT encoder, tile-based.
> > > - Gemma3: SigLIP encoder, fixed with pan and scan.
> > > - Molmo2: SigLIP2 encoder, multi-crop tiling
> > >
> > > This cross-family comparison reveals evidence for both architectural and training-data contributions.
> > >
> > > **Shared vulnerability points to the architecture.** All four families exhibit the same core pattern: spatial/resampling corruptions (upsample, elastic_transform) are universally the worst case. If the vulnerability were just training-data-driven, we would expect at least one family to escape this pattern, but none does.
> > >
> > > **Family-specific severity points to training.** InternVL3.5 is uniquely flip-sensitive (15.7% Flip+), Gemma3 is most fragile on reasoning (27.1% severe-fail on MMMU_Pro), and Molmo2 shows the purest degradation (Flip+/Flip- ratio of 4.2). These distinct profiles likely reflect differences in training data composition.
> > >
> > > **Domain-specific patterns suggest targeted training is needed.** OCR exhibits a distinct vulnerability fingerprint (elastic_transform dominates, not upsample), while general VQA is most affected by upsample. This suggests that a single augmentation strategy may not suffice; domain-specific training curricula may be needed for different task types.
> > >
> > > **Scaling alone is insufficient.** Within Qwen, scaling from 4B to 235B (22B active) improves mCE (50.0 vs 62.9), but worst-case drops remain 30.7pp. More data and parameters improve average robustness but do not eliminate the tail risk.
> > >
> > > **No current model uses spatial augmentation.** No evaluated model's technical report documents image augmentation during training, and Molmo2's open-source pipeline confirms no augmentation code. This is a clear gap.
> > >
> > > Fully isolating training vs. architecture would require controlled ablations (same data, different encoders, and vice versa). However, this would require a computationally intensive setup to post-train different VLMs under the same training data and augmentations. We identify this as important future work, and our benchmark provides the evaluation infrastructure to support it.
> > >
> > > **Roadmap for the field**
> > >
> > > Based on our findings, we propose four concrete directions (Section 6 of our paper):
> > > 1. Geometric data augmentation
> > > 2. Robustness-aware evaluation
> > > 3. Family-specific curricula
> > > 4. Visual reliance reporting
> > >
> > > **Clarifying the 4 families, attention maps, and failure cases**
> > >
> > > We will clarify the 4 vision encoder families in a detailed comparison table in the camera-ready version. We will also add attention map visualizations (comparing clean vs. corrupted inputs for upsample and elastic_transform) and failure case studies across task types (e.g., OCR under elastic_transform, relation reasoning under upsample) to provide more actionable insights.
> > >
> > > We hope we have addressed your concerns.

---

### Official Review · Reviewer_1Zoh · 2026-03-11

**Soundness:** 3
**Presentation:** 2
**Significance:** 3
**Originality:** 3
**Overall Recommendation:** 3
**Confidence:** 3

**Summary:**

This paper introduces VLM-RobustBench, a comprehensive benchmark evaluating the robustness of 11 state-of-the-art open-weight Vision Language Models  across MMMU-Pro and MMBench. Testing 49 image augmentations, the study reveals counter-intuitive findings: visual severity poorly predicts model difficulty. Mild spatial perturbations and trivial binary transforms often cause catastrophic accuracy drops, outperforming severe photometric distortions. The authors attribute this "spatial fragility" to the patch-based architecture of Vision Transformers, which are highly sensitive to resampling artifacts that disrupt spatial alignment. Furthermore, the analysis distinguishes between visually grounded tasks (MMBench) and those relying on language priors (MMMU-Pro). The paper advocates for integrating geometric data augmentation into training pipelines and establishing spatial robustness as a key evaluation metric to build more resilient VLMs.

**Compliance With Llm Reviewing Policy:**

Affirmed.

**Key Questions For Authors:**

Please refer to the weaknesses.

**Limitations:**

yes

**Strengths And Weaknesses:**

Strengths: The study uncovers counter-intuitive failure modes by revealing a mismatch between visual severity and model difficulty, showing that mild spatial perturbations are more detrimental than severe noise. It establishes a comprehensive evaluation framework with 133 configurations covering geometric, resampling, and photometric degradations. The authors provide deep theoretical insights by linking spatial fragility to the patch-based architecture of Vision Transformers and their sensitivity to interpolation artifacts. By introducing Visual Gain (VG) and Relative Corruption Error (RCE), the work effectively distinguishes between models relying on visual grounding versus language priors. Furthermore, it offers actionable development guidelines, proposing specific training strategies like geometric augmentation and new evaluation protocols to enhance VLM robustness.

Weaknesses
1. Generally, the robustness of VLMs to image corruptions is largely determined by training data and data augmentation strategies; however, the paper lacks an analysis of the impact of training data on VLM robustness.
2. The paper only evaluates VLMs with fewer than 30B parameters; therefore, it remains unknown whether the experimental results follow the same patterns for models exceeding 30B parameters.
3. After applying image corruptions to experimental samples, such as horizontal flipping, if the text prompt or ground truth involves content sensitive to left-right orientation, the ground truth should be correspondingly updated. Failing to do so may compromise the accuracy of the experimental results.

---

> ### Author Rebuttal · Authors · 2026-03-31
>
> Thank you for your comments and feedback. We would like to respond to the reviewer’s main concern below.
>
> >Generally, the robustness of VLMs to image corruptions is largely...
>
> We reviewed the available technical reports for all evaluated models and found that none document any image augmentation strategy during training; Molmo2's open-source data pipeline, which we inspected directly, contains no augmentation code either. This marks a departure from the ImageNet era, when geometric and photometric augmentations were standard practice.
> Instead, current VLMs appear to rely on the sheer diversity of large-scale web data to achieve robustness, yet our results show this is not enough for spatial perturbations. Our benchmark is designed precisely to surface such blind spots, without requiring access to proprietary training data.
>
> ---
> > The paper only evaluates VLMs with fewer than 30B parameters...
>
> We added **three new open-source models**, namely Qwen3-VL-235B (22 active), Qwen3-VL-32B, InternVL3.5-38B, and **one closed-source model**, GPT-5.4-mini, and ran the full 133-configuration evaluation on both datasets.
> **MMBench (Direct)**
>
> | Model | Baseline | Worst | Severe-Fail | Worst@Low | Benign@Low | VG | mRCE |
> |-------|----------|-------|-------------|-----------|------------|------|------|
> | Qwen3-VL-235B | 93.4 | 30.7 | 23.3 | 6.3 | 92.9 | 50.2 | 4.5 |
> | Qwen3-VL-32B | 92.4 | 32.4 | 5.3 | 8.1 | 81.0 | 49.2 | 5.4 |
> | InternVL3.5-38B | 90.5 | 27.2 | 3.0 | 6.1 | 85.7 | 46.8 | 4.5 |
> | GPT-5.4-mini | 87.7 | 31.8 | 6.8 | 11.7 | 85.7 | 46.3 | 5.9 |
>
> **MMMU-Pro (Direct)**
>
> | Model | Baseline | Worst | Severe-Fail | Worst@Low | Benign@Low | VG | mRCE |
> |-------|----------|-------|-------------|-----------|------------|------|------|
> | Qwen3-VL-235B | 46.0 | 20.7 | 7.5 | 13.6 | 95.2 | 13.6 | 9.4 |
> | Qwen3-VL-32B | 43.2 | 13.0 | 3.0 | 4.9 | 69.0 | 12.7 | 11.6 |
> | InternVL3.5-38B | 41.4 | 9.6 | 0.0 | 4.0 | 83.3 | 13.0 | 6.7 |
> | GPT-5.4-mini | 39.2 | 13.0 | 1.5 | 9.6 | 73.8 | 12.7 | 12.6 |
>
> mCE (normalized against reference): Qwen3-VL-235B **50.0**, Qwen3-VL-32B 58.6 / InternVL3.5-38B 66.4 / GPT-5.4-mini 85.9 (MMBench).
>
> Scaling improves mCE (Qwen3-VL-235B: 50.0 vs previous best 62.9), but worst-case drops remain substantial (235B: 30.7pp, 32B: 32.4pp, 38B: 27.2pp, GPT: 31.8pp). The vulnerability patterns observed in the paper (upsample/elastic_transform as worst cases, severity mismatch) hold consistently across all four new models.
>
> ---
>
> >After applying image corruptions to experimental samples,...
>
> This is a valid concern. To quantify it, we used **GPT-5.4 as a judge** to classify each question as spatially sensitive. The judge receives the image and the question text and determines whether answering requires knowledge of left-right or top-bottom orientation.
> Only **3-6%** of questions are inherently spatially sensitive (47/853 MMBench, 10/324 MMMU_Pro). Examples include "Which direction is the baby facing?" and map-based location questions.
>
> We recompute flip drops after excluding all sensitive questions:
>
> | Dataset | Flip | All Qs | Non-Sensitive Only | Drop Retained |
> |---------|------|--------|--------------------|---------------|
> | MMBench | H-flip | 7.1pp | 6.4pp | 90% |
> | MMBench | V-flip | 10.3pp | 10.0pp | 97% |
> | MMMU_Pro | H-flip | 3.9pp | 3.9pp | 100% |
> | MMMU_Pro | V-flip | 4.7pp | 4.7pp | 100% |
>
> 90-100% of the flip-induced drops remain after excluding all spatially sensitive questions. This confirms the drops are driven by model fragility to geometric transforms, not by ground-truth invalidation. We will add these results to the camera-ready.
>
> ---
>
> We appreciate the detailed comments and hope we have addressed most concerns.

---

### Official Review · Reviewer_qg8d · 2026-03-13

**Soundness:** 3
**Presentation:** 3
**Significance:** 3
**Originality:** 3
**Overall Recommendation:** 4
**Confidence:** 3

**Summary:**

This paper introduces VLM-RobustBench, an empirical benchmark designed to assess the robustness of Vision-Language Models (VLMs) against a wide array of visual corruptions. The benchmark applies 49 types of image augmentations—including noise, blur, weather, digital, geometric, and VLM-specific perturbations—across multiple severity levels to two standard multimodal evaluation datasets (MMBench and MMMU-Pro). The authors evaluate 11 recent open-weight VLMs and uncover several counter-intuitive findings. Most notably, they identify a "Spatial Fragility" phenomenon where VLMs are disproportionately sensitive to spatial and resampling corruptions (e.g., upsampling, elastic transformations, flipping) compared to severe photometric distortions. Additionally, they observe a "Severity Mismatch," indicating that visually mild perturbations can sometimes degrade performance more drastically than visually severe ones.

**Compliance With Llm Reviewing Policy:**

Affirmed.

**Key Questions For Authors:**

1. Spatial Tasks: How do these spatial corruptions impact tasks that strictly rely on high-resolution spatial details, such as Document VQA or visual grounding?

2. Dynamic Resolution: Could the extreme sensitivity to spatial/resampling changes be a side effect of the dynamic resolution and patching strategies used by models like Qwen3-VL and InternVL?

**Limitations:**

yes

**Strengths And Weaknesses:**

Strengths:
1. Insightful and Counter-intuitive Findings: The identification of "Spatial Fragility" and "Severity Mismatch" provides highly valuable insights for the VLM community. Highlighting that modern VLMs struggle with simple geometric/resampling changes despite handling severe photometric noise well exposes a critical vulnerability in current architectures.
2. Comprehensive Model Coverage: The evaluation is highly up-to-date, covering 11 recent and state-of-the-art open-weight VLMs spanning various architectures and parameter scales, making the findings broadly representative of the current field.

Weaknesses:
1. Limited Task Diversity for a "Comprehensive" Benchmark: While the title claims a "Comprehensive Benchmark," the base evaluation is limited to only two datasets (MMBench and MMMU-Pro). Given that spatial and resampling corruptions are the identified primary vulnerabilities, evaluating robustness on tasks inherently sensitive to high-resolution spatial details—such as OCR-centric document understanding (e.g., DocVQA, TextVQA) or fine-grained dense grounding (e.g., RefCOCO)—is essential to fully justify the "comprehensive" claim.
2. Lack of Deep Root-Cause Analysis: In Section 5.8, the authors hypothesize that the spatial fragility stems from the patch-based architecture of Vision Transformers (ViTs) underlying most VLMs. However, the paper does not provide empirical evidence to substantiate this claim. An ablation study investigating how different vision encoder designs (e.g., varying patch sizes, different tokenization granularities, or comparing with alternative architectures) react to these specific corruptions would significantly elevate the scientific depth of the paper.
3. Incremental Conceptual Novelty: Applying parameterized image corruptions (a la ImageNet-C) to evaluate model robustness is a well-established practice in computer vision. While migrating and adapting this paradigm to VLMs is highly useful and empirically solid, the core methodological novelty remains somewhat incremental.

---

> ### Author Rebuttal · Authors · 2026-03-31
>
> Thank you for your comments and feedback. We would like to respond to the reviewer’s main concern below.
> > 1. Limited Task Diversity for a "Comprehensive" Benchmark:
>
> While our benchmark uses two datasets, we covered diverse task categories via stratified sampling, as described in Section 4.2; we will clarify this further in the camera-ready version
> MMBench includes 20 fine-grained task types: **OCR (31 Qs), object localization (62 Qs), spatial reasoning (35 Qs), structured image-text understanding (56 Qs)**, action recognition, emotion, etc. MMMU_Pro spans **30 academic subjects across 6 domains**.
>
> Per-category analysis confirms spatial fragility across all task types (9 paper models avg + new Qwen3-VL-32B):
>
> | MMBench Category | 9-Model Avg | Qwen3-VL-32B |
> |-----------------|-------------|--------------|
> | | Clean / Worst | Clean / Worst |
> | relation_reasoning | 89.3% / 39.6pp | 96.6% / **43.7pp** |
> | finegrained_perception (instance) | 88.9% / 36.4pp | 92.9% / 36.2pp |
> | logic_reasoning | 83.8% / 28.4pp | 92.6% / **33.3pp** |
> | attribute_reasoning | 89.9% / 32.8pp | 93.5% / 29.0pp |
> | coarse_perception | 90.2% / 24.8pp | 90.8% / 22.9pp |
>
> Worst-case drops range from 23pp to 44pp across all categories, confirming that spatial fragility is not confined to a single task type. Notably, scaling to 32B parameters does not resolve this vulnerability: Qwen3-VL-32B actually suffers *larger* worst-case drops than the 9-model average on relation reasoning (43.7pp vs 39.6pp) and logic reasoning (33.3pp vs 28.4pp).
>
> **OCR-specific finding**: OCR is the most vulnerable category. Despite near-perfect clean accuracy (98.2% averaged across 9 models; 100% for Qwen3-VL-32B), worst-case drops are the largest of any task type: 52.0pp on average and 45.2pp for Qwen3-VL-32B. The dominant corruption also differs from general VQA: while upsample causes the worst degradation on most tasks, OCR is most affected by elastic_transform, which warps character geometry and destroys text readability.
>
>
> **MMMU_Pro**: Spatial fragility also holds on MMMU_Pro, where all 6 academic domains exhibit non-trivial worst-case drops (5.1pp in Tech & Engineering, 6.3pp in Science, 29pp in Art & Design), showing that spatial fragility generalises across both visually grounded and reasoning-heavy tasks.
>
> ---
>
> >2. Lack of Deep Root-Cause Analysis
>
> We acknowledge that controlled ablations (e.g., varying patch sizes on a fixed model) would be needed to fully isolate the mechanism and are beyond scope. However, our evaluation now spans **4 distinct vision encoder families** (Qwen-ViT, InternViT, SigLIP, standard ViT) with fundamentally different preprocessing (dynamic resolution, tile-based, fixed-resolution). All four exhibit the same core vulnerability pattern (upsample and elastic_transform as worst cases), yet diverge in family-specific severity (detailed in our response to R3 Q1). This cross-family consistency narrows the space of plausible explanations. We identify controlled ablations as an important direction for future work.
>
> >3. Incremental Conceptual Novelty
>
> We respectfully argue that the contribution goes beyond migrating ImageNet-C to VLMs, in both *scale* and *substance*.
> In terms of *scale*, our benchmark evaluates **49 corruptions (133 configurations)**, significantly more (3x+) than ImageNet-C's 15 corruptions. As Reviewer F8P6 also notes, our suite is "substantially more comprehensive than papers that test only a handful of ImageNet-C-style perturbations."
> In terms of *substance*, our framework surfaces three phenomena that are specific to VLMs and have no analogue in pure vision robustness evaluation:
>
> 1) Severity mismatch: Mild spatial perturbations cause more damage than severe photometric noise, contradicting ImageNet-C's monotonic severity-degradation relationship.
> 2) Visual Gain: VG reveals models relying on language priors rather than visual grounding, a failure mode absent in image-only classifiers.
> 3) Task-dependent vulnerability: The worst-case corruption differs by task type within a single model (elastic_transform for OCR vs. upsample for general VQA).
>
> > 1. Spatial Tasks
>
> See W1 above: OCR is the **most vulnerable** category (45-52pp worst-case drop), with elastic_transform as the dominant failure mode rather than upsample.
>
> ---
>
> > 2 Dynamic Resolution
>
> Our evaluation protocol controls for this: drops are measured relative to a clean baseline that passes through the **same** pipeline, so any resolution-related artifacts cancel out. Moreover, Molmo2 and Gemma3 use fixed-resolution preprocessing yet exhibit the same spatial fragility as Qwen3-VL (dynamic resolution) and InternVL3.5 (tile-based). The consistency across 4 fundamentally different preprocessing strategies rules out dynamic resolution as the driver.
>
> ---
> We appreciate the detailed comments and hope we have addressed most concerns.

---

> > ### Author Rebuttal · Reviewer_qg8d · 2026-04-01
> >
> > The authors’ rebuttal has addressed my concerns.

---

> > > ### Author Response · Authors · 2026-04-01
> > >
> > > We are glad that our responses have addressed your concerns.
> > > We would appreciate it if you could consider updating the overall score to reflect this, if you feel it is appropriate.

---

### Decision · Program_Chairs · 2026-04-30

**Decision:**

Accept (regular)

**Comment:**

This paper introduces VLM-RobustBench, a benchmark evaluating the robustness of vision-language models across augmentations and corruptions on MMBench and MMMU-Pro, revealing that current VLMs are more vulnerable to spatial and resampling corruptions (e.g., upsample, elastic_transform) than to visually severe photometric distortions, and that low-severity perturbations (e.g., glass_blur) can degrade performance more than high-severity ones. Reviewers recognized strengths such as comprehensive corruption suite, counter-intuitive findings on spatial fragility and severity mismatch, and broad model coverage, but raised concerns about limited task diversity (only two datasets MMBench and MMMU-Pro), absence of proprietary and larger models, and lack of root-cause analysis through controlled ablations. During the rebuttal, the authors added four new models including GPT-5.4-mini, Qwen3-VL-235B (22 active), Qwen3-VL-32B, and InternVL3.5-38B, confirming spatial fragility persists at scale and across proprietary models, provided per-category robustness breakdowns (OCR, object localization, spatial reasoning, structured image-text understanding), and addressed the ground-truth validity concern for spatial transforms (horizontal flip changes the answer to questions about direction). Reviewers F8P6 and qg8d were fully satisfied with the rebuttal, Reviewer nrEn was largely satisfied with the rebuttal pending camera-ready additions, while Reviewer 1Zoh did not acknowledge the rebuttal. The AC confirms that the concerns raised by reviewer 1Zoh have also been addressed by the rebuttal. Therefore, AC recommends acceptance contingent on incorporating the promised attention map visualizations and expanded analysis into the camera-ready.